# FYAI: A Fengyun Satellite-Based Dataset for Atmospheric Ice Water Path

Yifan Yang[1,2], Tingfeng Dou[1,2], Gaojie Xu[1,2], Rui Zhou[1,2], Bo Li[3], Letu Husi[4], Wenyu Wang[5], Cunde Xiao[6]

[1]College of Resources and Environment, University of Chinese Academy of Sciences, Beijing, China.

[2]State Key Laboratory of Earth System Numerical Modeling and Application, University of Chinese Academy of Sciences, Beijing, China

[3]Innovation Center for Fengyun Meteorological Satellite (FYSIC), Key Laboratory of Radiometric Calibration and Validation for Environmental Satellites, National Satellite Meteorological Center (National Center for Space Weather), China Meteorological Administration, Beijing 100081, China

[4]State Key Laboratory of Remote Sensing and Digital Earth, Aerospace Information Research Institute, Chinese Academy of Sciences, Beijing, 100101, China

[5]Key Laboratory of Microwave Remote Sensing, National Space Science Center, Chinese Academy of Sciences, Beijing 100190, China

[6]State Key Laboratory of Earth Surface Processes and Disaster Risk Reduction, Beijing Normal University, Beijing, 100875, China

*Correspondence to*: Tingfeng Dou (doutf@ucas.ac.cn)

**Abstract.** This study introduces FYAI, a global, long-term atmospheric Ice Water Path (IWP) and Suspended Ice Water Path (SIWP) dataset spanning 2010-2024, derived from passive microwave observations (MWHS-I/II) onboard China's Fengyun-3 series satellites. The dataset is generated using a machine learning framework featuring a lightweight multilayer perceptron architecture enhanced with gated residual units. This design robustly handles the inherent uncertainties in satellite brightness temperatures and the spatial mismatch between passive microwave footprints and active radar/lidar training data. By establishing rigorous spatiotemporal collocation with CloudSat 2C-ICE products, FYAI provides two operational product levels adhering to standard Earth observation data processing definitions: (1) Level-2 (L2) products, offering instantaneous orbital-resolution IWP and SIWP at a nominal 15 km nadir resolution (2010-2024); and (2) Level-3 (L3) products, comprising monthly global gridded composites at 1° × 1° resolution (2010-2024). FYAI bridges the gap between instantaneous pixel-level precision and broad spatiotemporal coverage, offering a comprehensive, decadal-scale record of global atmospheric ice content. This dataset, specifically designed to support long-term climate analysis and model validation, is openly available in netCDF4 format for community use.

## 1 Introduction

Ice crystals play a pivotal role in cloud and precipitation processes, thereby significantly modulating the hydrological cycle, thermodynamics, and radiative transfer (Gultepe et al., 2017). Consequently, the reliable quantification of atmospheric ice content is critical for elucidating latent heat distribution and precipitation mechanisms (Amell et al., 2022). The primary metric used to describe this ice content is the ice water path (IWP), defined as the vertical integral of the ice water content (IWC). IWP is composed of both suspended ice and falling ice (also referred to as precipitation ice), although the criteria distinguishing these components remain ill-defined (Eliasson et al., 2011; Waliser et al., 2009). However, current climate models exhibit widespread inconsistencies and pronounced spatial heterogeneity in simulating IWP (Eriksson et al., 2025; Wang, 2022). Indeed, as highlighted in the Intergovernmental Panel on Climate Change Sixth Assessment Report (IPCC AR6), these cloud and precipitation processes remain primary sources of uncertainty in climate modeling and projections (IPCC, 2023). This underscores the critical need for high-quality observational constraints on atmospheric ice (Holl et al., 2014).

From an observational perspective, space-based remote sensing is the primary means of providing global IWP data, yet existing products face limitations. Visible and infrared sensors, such as MODIS and AIRS, have provided valuable long-term records. However, their measurements are often constrained by signal saturation in optically thick clouds, and they are primarily sensitive to upper cloud layers rather than probing the full depth of deep convective systems (Eliasson et al., 2011). Conversely, limb sounders like the Microwave Limb Sounder (MLS), while offering vertical profiles, are constrained by extremely sparse horizontal sampling, making them unsuitable for continuous regional monitoring (Wu et al., 2006). Active sensors (e.g., CloudSat/CALIPSO) offer high accuracy but represent only a "needle-thin" curtain of the atmosphere (Delanoë and Hogan, 2010; Hong and Liu, 2015). Consequently, passive microwave instruments remain the optimal solution for retrieving large-scale, long-term, and all-weather IWP data due to their ability to penetrate dense clouds and interact directly with ice mass (Evans and Stephens, 1995; Wu et al., 2008).

Currently, microwave humidity sounders operating below 200 GHz (e.g., AMSU-B, MHS) are standard for ice detection. However, despite carrying Microwave Humidity Sounder (MWHS), the potential of

China's Fengyun-3 (FY-3) series satellites remains largely untapped in producing global climate
datasets. The FY-3 series offers a unique advantage unmatched by other operational systems: a
complete three-orbit constellation comprising morning (FY-3A/C/F), afternoon (FY-3B/D), and the
distinct dawn-dusk (FY-3E) orbit satellites (An et al., 2023; Tan et al., 2019; Wang et al., 2022). This
configuration allows for substantially improved temporal sampling, filling critical gaps in the diurnal
cycle of IWP that are missed by sun-synchronous satellites restricted to fixed crossing times,
particularly with the inclusion of FY-3E observations starting in 2023. By leveraging this 15-year
continuous archive (2010-2024), there is an opportunity to construct a coherent, long-term IWP climate
data record that overcomes the spatiotemporal limitations of existing datasets.

While traditional physical retrieval methods offer interpretability, they rely heavily on complex
scattering databases and microphysical assumptions (e.g., particle shape and size distribution) that are
often difficult to constrain globally. (Letu et al., 2016, 2020). Machine learning (ML) has emerged as a
powerful alternative for handling the non-linear relationships in passive microwave retrieval. Previous
efforts, such as SPARE-ICE (Holl et al., 2014) or geostationary retrievals (Amell et al., 2022, 2024;
Tana et al., 2025), have demonstrated the efficacy of NN-based approaches. Similarly, recent studies
involving co-authors of this paper have explored ML applications on IWP retrieval using polar-orbiting
FY-3 satellites (Wang et al., 2022, 2024). However, a dedicated, long-term IWP dataset derived
specifically from the advanced capabilities of the FY-3 constellation—which also incorporates a
distinction between total ice and suspended ice—is currently absent from the community.

To address these gaps, this study presents "FYAI" (Fengyun Satellite-Based Dataset for Atmospheric
Ice Water Path), a novel global dataset generated using a NN-based framework. By training on 2C-ICE
active remote sensing data and applying it to the MWHS-I/II records from the entire FY-3 family, FYAI
provides a seamless 15-year record (2010-2024) of both Level-2 (L2) and Level-3 (L3) monthly
gridded IWP. A unique feature of FYAI, achieved by integrating 2B-CLDCLASS product, is its ability
to provide a separate product specifically for Suspended IWP (SIWP), distinguishing it from falling ice.
This distinction offers additional observational constraints for climate models. FYAI offers a unique
combination of all-sky capability, dense spatial coverage, and the first-ever inclusion of dawn-dusk
microwave observations, offering new insights into the global atmospheric ice content.
**2 Data**
**2.1 Input data**
The primary passive microwave instruments utilized in this study are the MWHS-I and MWHS-II,
onboard China's second-generation polar-orbiting FY-3 series meteorological satellites. The MWHS-I
is carried on the initial batch of these satellites (FY-3A and FY-3B). The MWHS-II represents a
significant upgrade and was deployed in two successive batches: the first batch aboard the second
satellite group (FY-3C, FY-3D), and the second batch aboard the third group (FY-3E, FY-3F). It
expands the channel count from 5 to 15, adding new oxygen absorption channels near 118.75 GHz and
a window channel at 89 GHz (Wang et al., 2024). Both MWHS-I and MWHS-II operate as cross-track
scanners. The MWHS-I offers a nadir resolution of approximately 15 km across all its channels. For the
MWHS-II, all channels also have a nadir resolution of about 15 km, with the exception of the 89 GHz
and 118 GHz channels, which have a coarser nadir resolution of approximately 25 km. Detailed
channel specifications, instrument parameters, and the data temporal coverage for each satellite are
provided in Supplementary Tables S1-S4.
For input into our retrieval model, we selected not only the Level-1 (L1) brightness temperature data
from these instruments but also a suite of auxiliary geographical and geometric parameters. These
additional features include the Digital Elevation Model (DEM), solar zenith angle, satellite zenith angle,
land-sea mask etc. A comprehensive list of all input variables is presented in Table 1.
**Table 1 All input variables**

|  | Brightness Temperature data | Auxiliary data |
|---|---|---|
| Model for MWHS | $BT_1$ (150 GHz (V)), $BT_2$ (150 GHz (H)), $BT_3$ (183.31±1GHz), $BT_4$ (183.31±3GHz), $BT_5$ (183.31±7GHz), | SensorAzimuth, SensorZenith, SolarAzimuth, SolarZenith, LandSeaMask, DEM, Longitude, Latitude |
| Model for MWHS-II | $BT_1$ (89GHz), $BT_{11}$ (183.31±1GHz), $BT_{12}$ (183.31±1.8GHz), $BT_{13}$ (183.31±3GHz), $BT_{14}$ (183.31±4.5GHz), $BT_{15}$ (183.31±7GHz) | SensorAzimuth, SensorZenith, SolarAzimuth, SolarZenith, LandSeaMask, LandCover, DEM, Longitude, Latitude |

**2.2 Reference data**

**2.2.1 2C-ICE**

The CloudSat and CALIPSO ice cloud property product (2C-ICE) is developed by synergistically integrating measurements from the CloudSat Cloud Profile Radar (CPR) and the CALIPSO CALIOP lidar. Specifically, it utilizes CPR radar reflectivity (from the 2B-GEOPROF dataset) alongside CALIOP attenuated backscatter at 532 nm. By combining the penetration capability of the radar with the high sensitivity of the lidar to tenuous ice, this joint approach effectively overcomes the limitations of single-instrument retrievals, yielding IWC estimates with enhanced accuracy (Deng et al. 2010). The base CPR data provides vertical profiles at a 240 m resolution with a 1.4 km × 1.8 km footprint. In this work, the 2C-ICE product is specifically employed to be the IWP reference value.

**2.2.2 2B-CLDCLASS**

The 2B-CLDCLASS product, based on CloudSat CPR observations, utilizes a multidimensional approach to categorize clouds with high precision. The classification framework integrates key parameters, including hydrometeor dimensions (vertical/horizontal scales) and the maximum radar reflectivity factor (Ze), alongside crucial ancillary data such as precipitation flags and ECMWF temperature profiles, which aid in phase determination (Sassen and Wang, 2008). While enabling robust cloud climatology studies, in this work, the 2B-CLDCLASS product is specifically employed to distinguish and extract the SIWP component from the IWP.

**2.3 Validation data**

To ensure comprehensive evaluation, multiple validation datasets are utilized alongside 2C-ICE. These include satellite-derived retrievals from active and passive remote sensing instruments, as well as independent reanalysis products.

**2.3.1 DARDAR (raDAR/liDAR) IWP**

DARDAR (raDAR/liDAR) is a synergistic ice-cloud retrieval that combines CloudSat radar and CALIPSO lidar measurements within a variational framework to yield profiles of extinction coefficient, ice water content and effective radius (Re) (Delanoë and Hogan, 2008, 2010; Hogan et al., 2006). The algorithm adopts the "unified" particle-size distribution of Field et al. (2005) and employs in-situ-derived mass–and area–dimension relations for non-spherical ice particles (Brown and Francis,

140   1995; Li et al., 2012).

**2.3.2 CCIC IWP**

The Chalmers Cloud Ice Climatology (CCIC) is a long-term climate data record of global Total Ice Water Path (TIWP). It is generated by a deep learning model using geostationary satellite infrared window channel observations and provides continuous, all-sky (day and night) TIWP estimates from 1983 to the present within 70°S-70°N, whuch has been demonstrated to agree well with other in-situ and active radar observations (Amell et al., 2024; Pfreundschuh et al., 2025).

**2.3.3 MODIS and VIIRS IWP**

This study utilizes operational IWP data derived from MODIS and VIIRS instruments, obtained through the CERES SSF1deg product suite.

The IWP is retrieved via a bispectral algorithm from imager radiances and represents the total column ice mass. The native high-resolution retrievals are aggregated to CERES footprints and subsequently averaged onto a 1° global grid. Daily and monthly means are generated after temporal interpolation of instantaneous values (Platnick et al., 2017).

**2.3.4 ERA5 IWP**

ERA5 is the fifth-generation global atmospheric reanalysis from the European Centre for Medium-Range Weather Forecasts (ECMWF). It provides globally complete, hourly estimates of atmospheric variables from 1940 onward at a horizontal resolution of 0.25°. The dataset is produced using a fixed version of the ECMWF's Integrated Forecasting System (CY41R2) and a 4D-Var assimilation system, which incorporates over 200 diverse observation sources to ensure physical consistency (Hersbach et al., 2020). In this study, the ERA5 variable "Total column cloud ice water" is used as SIWP, while the sum of "Total column cloud ice water" and "Total column snow water" represents the total IWP.

**3 Methodology**

**3.1 Preprocessing**

**3.1.1 Quality control**

The L1 data from the Fengyun series satellites include quality-related flags. For the MWHS-II data, three quality flags are provided: the scan line preprocessing quality flag (QA_Scan_Flag), the channel data integrity quality flag (QA_Ch_Flag), and the observed brightness temperature quality score (QA_Score). QA_Scan_Flag is an integer ranging from 0-12113, where 0 indicates successful preprocessing of the scan line. QA_Ch_Flag is a 16-bit binary code stored as an integer between 0 and 65534, with 0 indicating complete channel data. QA_Score ranges from 0 to 100, with higher values indicating better brightness temperature quality. This study sets the following quality thresholds: QA_Scan_Flag = 0, QA_Ch_Flag = 0, and QA_Score ≥ 90. For the MWHS-I data, the following quality flags are similarly provided: calibration quality flag (cal_qc), pixel quality flag (pixel_qc), and scan line quality flag (scnlin_qc). All three flags are integers ranging from 0-65535. We exclusively select data points where all three flags equal 0. Additionally, for the 2C-ICE product, we excluded data points where the 'Data_quality' variable was non-zero, as a value of 0 indicates good data quality.

**3.2 Collocations**

To meet the requirements of machine learning algorithms, passive instrument observations must be spatiotemporally matched with reference data. FY-3D and CloudSat are both satellites in afternoon orbits. FY-3D crosses the equator at approximately 2:00 PM local time, while CloudSat crosses at 1:30 PM. Due to CloudSat's orbital drift during operation, the time difference between it and FY-3D is mostly within 15 minutes. Consequently, temporal matching is straightforward, and a 15-minute time window was selected to account for typical convective system time scales.

Spatially, matching is more complex because MWHS-II has a coarser resolution than 2C-ICE, resulting in multiple 2C-ICE pixels falling within a single MWHS-II field of view (FOV). Based on previous studies (Holl et al., 2010; Wang et al., 2022), two criteria were initially adopted to ensure sufficient representativeness and homogeneity of the 2C-ICE pixels within each MWHS-II FOV: (1) at least nine 2C-ICE pixels must lie within a 7.5 km radius of the MWHS-II FOV center, and (2) the coefficient of variation (standard deviation divided by the mean) of these 2C-ICE pixels must be less than 0.6.


However, two critical limitations regarding this spatial matching approach must be acknowledged. First,
using a fixed 7.5 km distance threshold is imprecise because MWHS-II spatial resolution varies by
frequency: approximately 15 km at 150/183 GHz, but 25 km at 89/118 GHz. Since channels near 118
GHz are not included in our model input, only the 89 GHz channel differs in resolution from the others.
Although the 89 GHz channel has a coarser resolution (25 km) and is crucial for IWP retrieval  (Wang
et al., 2024), we prioritized the matching accuracy for the 183 GHz channels (15 km), which constitute
the majority of the input features. Therefore, the 7.5 km threshold is a compromise to ensure the
highest fidelity for the sounding channels, despite the partial spatial mismatch at 89 GHz. Second,
MWHS instruments are cross-track scanners, meaning their spatial resolution degrades as the scan
angle increases away from nadir (Fig. S1). The stated resolutions of 15/25 km represent the nadir
resolution (the theoretical maximum). This further indicates that using a fixed 7.5 km threshold across
the entire swath is not entirely accurate. While we plan to introduce a scan-angle-dependent variable
threshold in future updates, the fixed 7.5 km threshold was retained in the current version to maintain
algorithmic simplicity and consistency across the swath matched with the nadir resolution baseline.

Ultimately, using FY-3D data from October 2018 to October 2020, we generated a dataset containing
2,667,945 matched points. For the MWHS-I instrument, FY-3B is also an afternoon satellite with an
ascending node local time of 1:40 PM. We thus used its data from December 2010 to April 2011 and
matched them with corresponding 2C-ICE data following the same criteria applied for MWHS-II. This
process yielded 426,761 matched points. Both the MWHS-I and MWHS-II datasets were then split into
training and testing sets. Subsequently, the training set was further divided, with 80% used for model
training and the remaining 20% reserved for validation.

The calibration process for the SIWP training dataset followed an approach similar to that used for the
IWP dataset. Based on the FLAG methodology described by Li et al. (2012), we isolated the suspended
component of the ice water path. This involved applying strict filtering criteria: all retrievals identified
as surface precipitation were discarded. Furthermore, to minimize convective influence, we excluded
data points classified as 'deep convection' or 'cumulus' according to the 2B-CLDCLASS product.
Similarly, the final dataset consisted of 2,667,945 matched points for MWHS-II and 426,761 matched
points for MWHS-I.

**3.3 Postprocessing**
The L2 IWP product maintains a native spatial resolution of nominal 15 km at nadir. To support
climatological analysis, we generate monthly L3 products on a uniform $1° \times 1°$ global grid. This is
achieved by resampling and averaging all available L2 data points within each grid cell for each
calendar month.

**3.4 IWP retrieval algorithm**
To retrieve IWP from passive microwave remote sensing observations, we developed a NN-based
model built upon the framework of Quantile Regression Neural Networks (QRNNs). QRNNs synergize
the non-linear representation learning capabilities of neural networks with the statistical framework of
quantile regression. Unlike traditional regression models that estimate only the conditional mean of a
response variable, QRNNs are designed to estimate multiple conditional quantiles of the target
distribution simultaneously. This approach provides a comprehensive probabilistic view of the
prediction, quantifying the aleatoric uncertainty inherent in the data, which is particularly valuable in
remote sensing retrievals where robust uncertainty assessment is crucial. Previous studies have
demonstrated QRNNs to be a high-performance and readily deployable model in this field (Amell et al.,
2022; Pfreundschuh et al., 2018; Wang et al., 2024). Furthermore, to enhance model performance, we
implemented a deep residual network architecture combined with attention mechanisms (He et al.,
2016; Vaswani et al., 2017). This design allows the model to automatically focus on the most critical
feature channels in the input satellite data while maintaining high training stability. To enable the
prediction of this uncertainty range, our model employs the specialized Quantile Loss, also known as
the Pinball Loss, instead of the traditional Mean Squared Error (MSE) loss function. The formula for
the Quantile Loss is expressed as follows:
$$L_{\tau(x_\tau,x)} = \begin{cases} \tau|x - x_\tau| & x_\tau \leq x \\ (1 - \tau)|x - x_\tau| & \text{otherwise} \end{cases} \tag{1}$$
$$x_\tau = \inf\{x: F(x) \geq \tau\} \tag{2}$$
$$L(x) = \frac{1}{N}\sum_{i=0}^{N} L_{\tau_i}(\widehat{x_i}, x) \tag{3}$$

Based on the fundamental assumption in deep learning that the training set, test set, and inference data
are independent and identically distributed (i.i.d.), we calibrated our point estimation strategy using the
test set statistics. Specifically, the deterministic point estimate was defined as the quantile associated
with the mode of the optimal quantile distribution, calculated using 50 bins on the test set.
Consequently, the optimal quantile was determined to be 47.87% for the MWHS-I model and 40% for
the MWHS-II model. Additionally, the 5th and 95th percentiles were employed to define the
uncertainty bounds for the IWP estimates. The matched dataset is partitioned into training and
validation subsets. Prior to model training, the IWP reference values within the training set are
log-transformed. To handle zero values in this transformation, they are replaced with a small positive
value of $1\times10^{-6}$. Analogous procedures were applied to the SIWP retrieval model. The specific structure
of the model is shown in Figure 1, and the detailed hyperparameters are listed in Table S5.

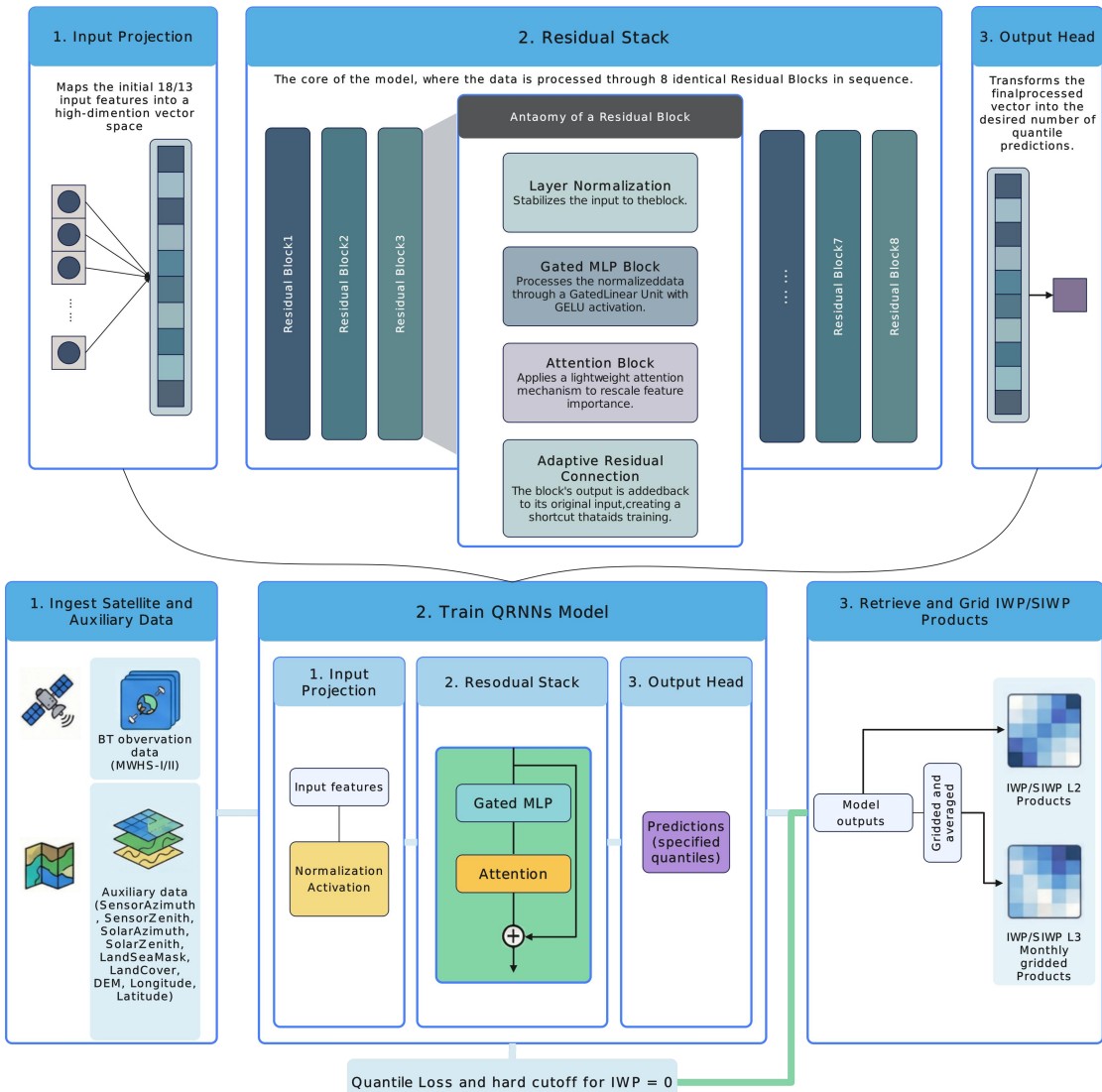

**Figure 1: Structural diagram of the QRNN model and flowchart of the retrieval algorithm.**

**3.5 Evaluation metrics**

The performance of the QRNN model in retrieving IWP is evaluated via the root mean square error (RMSE) and Pearson correlation coefficient (R), which are calculated as follows:

$$\text{RMSE} = \sqrt{\frac{1}{N}\sum_{i=1}^{N}\left(y_{\text{pred,i}} - y_{\text{ref,i}}\right)^2} \tag{4}$$

$$R = \frac{\frac{1}{N}\sum_{i=1}^{N}\left(y_{\text{pred,i}} - \overline{y_{\text{pred}}}\right)\left(y_{\text{ref,i}} - \overline{y_{\text{ref}}}\right)}{\sigma_{\text{pred}}\sigma_{\text{ref}}} \tag{5}$$

Here, $y_{\text{pred}}$ and $y_{\text{ref}}$ represent the model predictions and reference values, respectively, whereas $\sigma_{\text{pred}}$ and $\sigma_{\text{ref}}$ are the standard deviations.

For low IWP values regime detection, performance is evaluated via a confusion matrix M, with metrics
including FAR and CSI, defined as:
$$M = \begin{pmatrix} TP & FP \\ FN & TN \end{pmatrix} \quad (6)$$
True positives (TP) correspond to cases where both MWHS-I/II and CloudSat detect a low-IWP regime,
whereas true negatives (TN) occur when neither of them identifies such a regime. False positives (FP)
arise when MWHS-I/II detects a low-IWP regime that CloudSat does not confirm, and false negatives
(FN) occur when CloudSat identifies a low-IWP regime that MWHS-I/II fails to detect.
$$FAR = FP / (TP + FP) \quad (7)$$
$$CSI = TP / (TP + FN + FP) \quad (8)$$
**4 Data Records**
We have ultimately generated L2 IWP and SIWP products, as well as monthly gridded L3 IWP and
SIWP products, based on MWHS-I L1 data from the FY-3A/B satellites and MWHS-II L1 data from
the FY-3C/D/E/F satellites. The L2 products have a nadir spatial resolution of 15 km, while the L3
products are provided on a $1° \times 1°$ grid.

The    L2    products    adhere    to    the    file    naming    convention
"FY3X_MWHSX_GBAL_L2_YYYYMMDD_HHMM_015KM_FYAI.nc", where "YYYYMMDD"
and "HHMM" denote the date and start time (UTC) of the observation, respectively. Correspondingly,
the L3 gridded products are designated as "FY3X_L3 _Gridded_YYYY-YYYY_FYAI.nc".
Additionally, for the L3 products derived from FY-3E and FY-3F, the naming convention distinguishes
orbital    direction,    taking    the    form    "FY3X_L3_Gridded_YYYY-YYYY_FYAI_ascend.nc"    or
"FY3X_L3_Gridded_YYYY-YYYY_FYAI_descend.nc", where "ascend" and "descend" denote the
ascending and descending orbits, respectively. Detailed variable specifications for both product levels
are provided in Table 2, while the internal data structure and organization are visually depicted in
Figure 2. The temporal coverage for these datasets extends from 2010 to 2024.

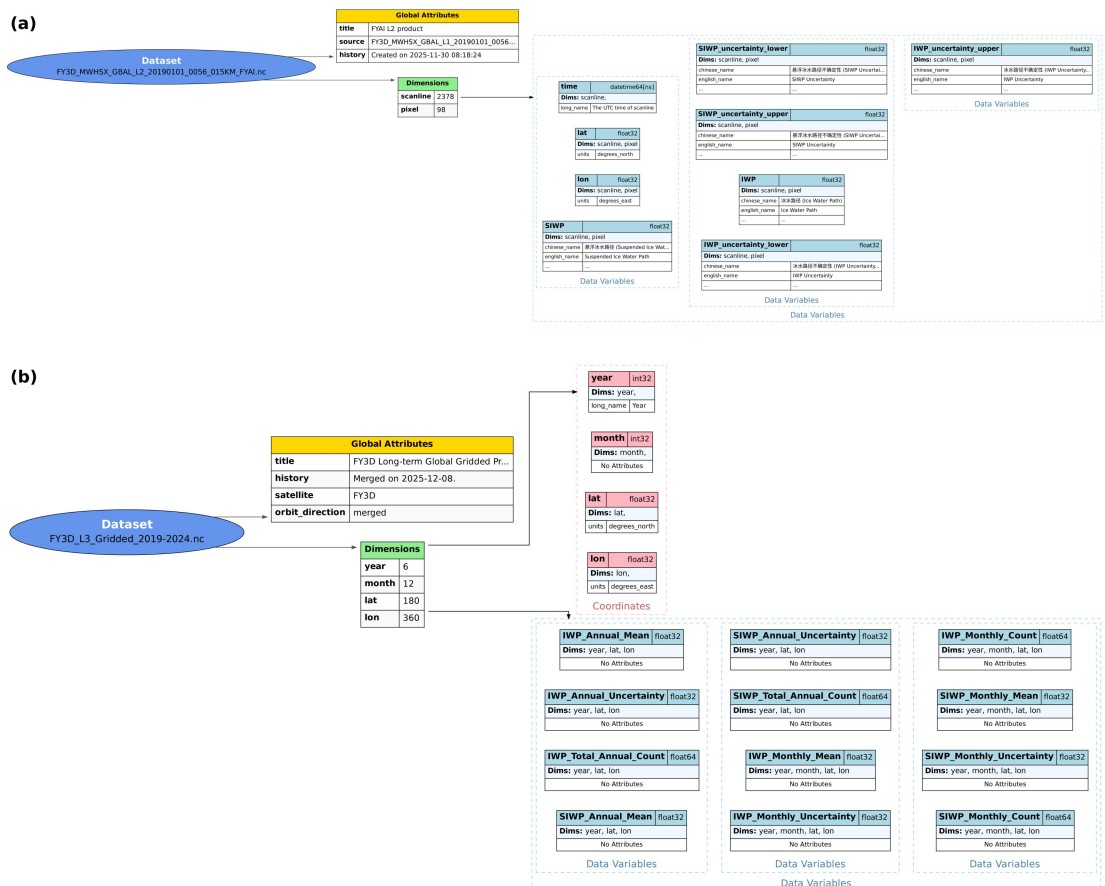

**Figure 2: Schematic of the data file structure: (a) L2 data file structure; (b) L3 data file structure.**

Figure 3 shows the monthly count of FY-3 L1 data inputs to the model. Due to operational anomalies, hardware upgrades, and other mission-related factors, data availability dropped below 50% in certain months. The 50% data-availability criterion is not meant as a benchmark for climate-grade accuracy; whether it suffices depends on the study's objectives and the natural variability of the target region (Bertrand et al., 2024; Kotarba et al., 2021). Nevertheless, we recommend that users exercise caution when utilizing data from months where availability falls below 50%.

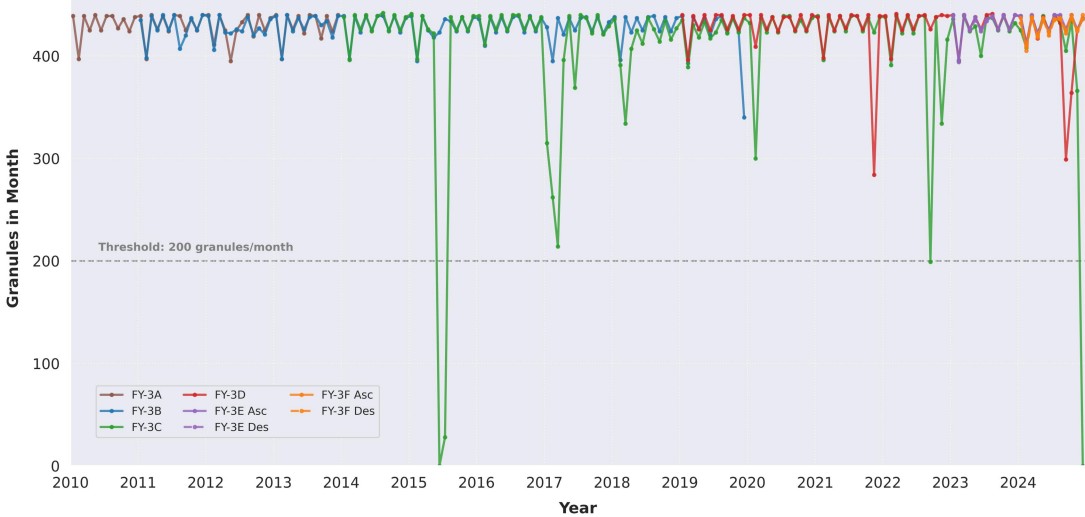


**Figure 3: MWHS-I and MWHS-II L1 data availability onboard the FY-3 series satellites.**


**Table 2: Data variables in FYAI L2 and L3 products.**

|  | Variable Name | Dimensions | Type | Description |
|---|---|---|---|---|
| L2 product | IWP | (scanline, pixel) | float32 | Ice Water Path |
|  | SIWP | (scanline, pixel) | float32 | Suspended Ice Water Path |
|  | IWP_uncertainty_upper | (scanline, pixel) | float32 | 95th quantile value of Ice Water Path |
|  | IWP_uncertainty_lower | (scanline, pixel) | float32 | 5th quantile value of Ice Water Path |
|  | SIWP_uncertainty_upper | (scanline, pixel) | float32 | 95th quantile value of Suspended Ice Water Path |
|  | SIWP_uncertainty_lower | (scanline, pixel) | float32 | 5th quantile value of Suspended Ice Water Path |
|  | lon | (scanline, pixel) | float32 | Longitude |
|  | lat | (scanline, pixel) | float32 | Latitude |
|  | time | (scanline) | datetime64 | The UTC time of scanline |
| L3 product | IWP_Annual_Mean | (year, lat, lon) | float32 | Annual Mean Ice Water Path from L2 observations |
|  | IWP_Annual_Uncertainty | (year, lat, lon) | float32 | Uncertainty (Standard Error of the Mean , SEM) of Annual Ice Water Path |

| | | | |
|---|---|---|---|
| IWP_Total_Annual_Count | (year, lat, lon) | float64 | Total number of valid L2 Ice Water Patobservations |
| SIWP_Annual_Mean | (year, lat, lon) | float32 | Annual Mean Suspended Ice Water Path from L2 observations |
| SIWP_Annual_Uncertainty | (year, lat, lon) | float32 | Uncertainty (SEM) of Annual Suspended Ice Water Path |
| SIWP_Total_Annual_Count | (year, lat, lon) | float64 | Total number of valid L2 Suspended Ice Water Path observations |
| IWP_Monthly_Mean | (year, month, lat, lon) | float32 | Monthly Mean Ice Water Path |
| IWP_Monthly_Uncertainty | (year, month, lat, lon) | float32 | Uncertainty (SEM) of Monthly Ice Water Path |
| IWP_Monthly_Count | (year, month, lat, lon) | float64 | Number of valid L2 Ice Water Path observations per month |
| SIWP_Monthly_Mean | (month, lat, lon) | float32 | Monthly Mean Suspended Ice Water Path |
| SIWP_Monthly_Uncertainty | (month, lat, lon) | float32 | Uncertainty (SEM) of Monthly Suspended Ice Water Path |
| SIWP_Monthly_Count | (year, month, lat, lon) | float64 | Number of valid L2 Suspended Ice Water Path observations per month |
| lon | (lon,) | float32 | Longitude |
| lat | (lat,) | float32 | Latitude |
| month | (month,) | int32 | Month of year |
| year | (year) | Int32 | year |


## 5 IWP retrieval performance

It is important to acknowledge that since the QRNN model was trained and tested based on the 2C-ICE
dataset, it inevitably inherits the systematic biases of the 2C-ICE product. Previous studies have
indicated that assumptions regarding the lidar ratio, particle size distribution (PSD), and particle shape
in the 2C-ICE retrieval algorithm introduce systematic uncertainties. Comparisons with in-situ
observations suggest an uncertainty of approximately 30% in 2C-ICE retrieved IWC (Deng et al., 2010,

321    2013).


Figure 4 illustrates the comparison of IWP retrieval performance between the two satellite sensors. In
terms of quantitative regression metrics, the model performance on FY-3D is significantly superior to
that on FY-3B. Specifically, the scatter plot for FY-3D (Figure 4a) shows a high consistency between
predicted and reference values, with a correlation coefficient (R) of 0.833 and a RMSE of 450.78 g/m².
In contrast, the scatter distribution for FY-3B (Figure 4d) is more dispersed, yielding a lower R of
0.620 and a larger RMSE (871.40 g/m²). This disparity highlights the substantial contribution of the
rich channel information provided by MWHS-II to the quantitative retrieval of IWP.

Regarding statistical distribution, we analyzed both the Quantile-Quantile (Q-Q) plots (Figure 4b and
Figure 4e) and the Probability Density Functions (PDFs, Figure 5) based on an independent test dataset.
As shown in the PDF analysis, the retrieved IWP distribution exhibits remarkable agreement with the
reference distribution across nearly six orders of magnitude (ranging from $10^{-2}$ to $10^4$ g/m²). This
confirms that the model successfully reproduces the climatological statistics without suffering from
significant mean-reversion. Both the PDFs and Q-Q plots indicate that the model robustly captures the
data distribution characteristics. Critically, given the global mean IWP of approximately 100 g/m² (Xu
et al., 2022), the model maintains robust performance across predominant atmospheric conditions.
However, deviations are observed in the extremely low-value region in the Q-Q plots. This is likely
attributable to the inherent physical limitations of passive microwave remote sensing, which is
sensitive to large scatterers (e.g., snowflakes) but lacks sensitivity to small ice crystals.

To further investigate model performance in the low-IWP value range, we performed a binary
classification assessment on the test set using a threshold of 0.5 g/m². The results (Figure 4c and Figure
4f) reveal distinct characteristics for the two sensors. Although FY-3D achieves higher quantitative
retrieval accuracy, its confusion matrix (Figure 4c) indicates a relatively high False Alarm Ratio (FAR
= 0.76) and a lower Critical Success Index (CSI = 0.23). This is primarily due to a large number of
background pixels (low values) being misclassified as exceeding the threshold (FP = 257,474).
Conversely, while FY-3B (Figure 4f) has lower regression accuracy, it exhibits a better balance in

classification metrics, with a lower FAR (0.51) and a relatively higher CSI (0.48). While this difference

may be partially influenced by the varying sample sizes in the test sets, it suggests that the FY-3D

model, while accurate in estimating IWP magnitude, tends to be over-sensitive at the boundary between

weak signals and background noise.

The performance analysis for SIWP yields similar conclusions to those for IWP and is detailed in the

Supplementary Material (Fig. S2, Text S2).

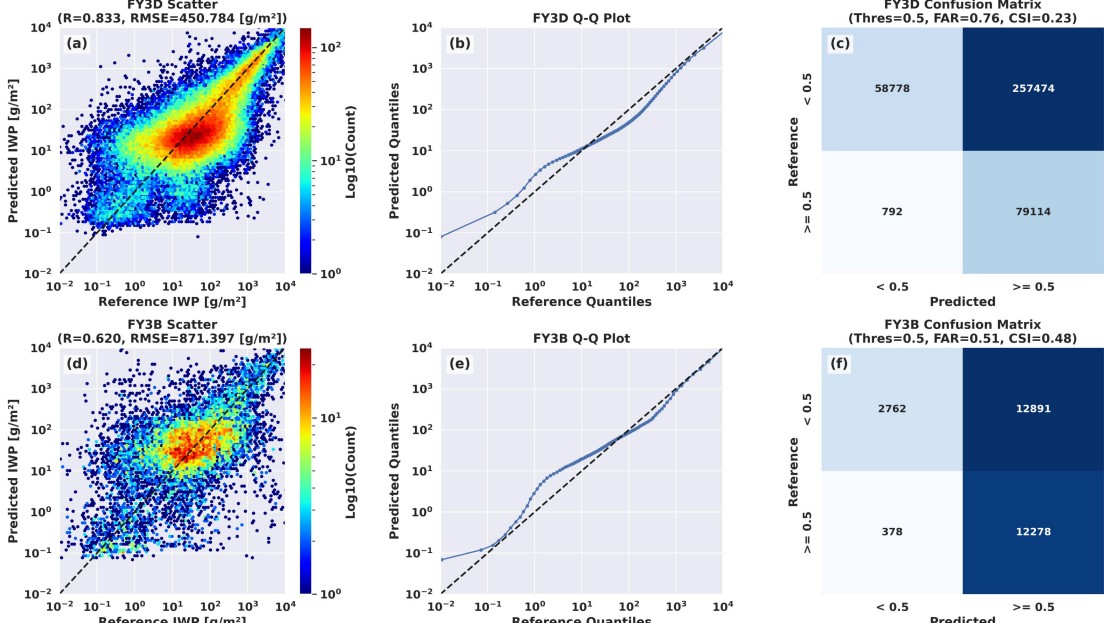

**Figure 4: Performance metrics of the QRNN model on the IWP test dataset. (a) scatter plot of mode-retrieved IWP values versus reference values on MWHS-II; (b) Q-Q plot of predicted values versus reference values on MWHS-II; (c) confusion matrix for MWHS-II using an IWP threshold of 0.5 g/m²; (d) analogous to (a) but for MWHS-I; (e) analogous to (b) but for MWHS-I; (f) analogous to (c) but for MWHS-I.**

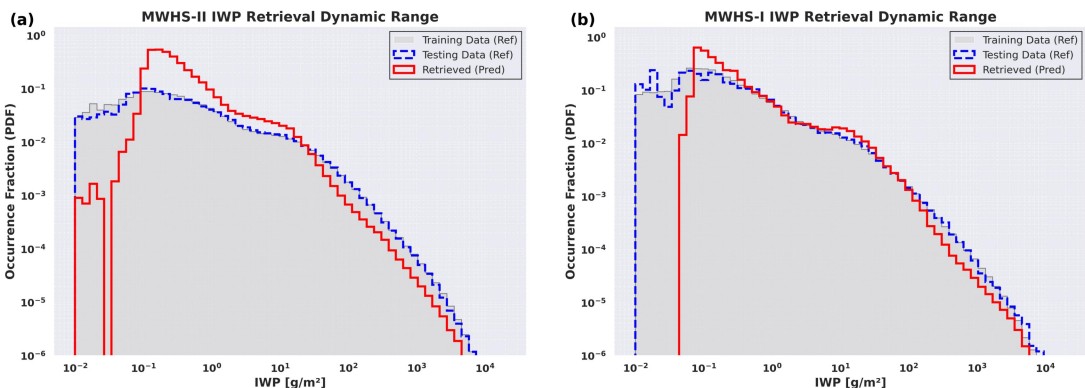

**Figure 5: Probability Density Functions (PDFs) of Ice Water Path (IWP) for the training dataset, testing dataset, and model retrievals. (a) FY-3D (MWHS-II model); (b) FY-3B (MWHS-I model). The histograms are calculated using logarithmically spaced bins to capture the wide dynamic range.**

## 6 Product validation

### 6.1 Typhoon events

Figure 6 presents the FYAI L2 IWP retrievals, alongside IWP estimates from the 2C-ICE product, the CCIC dataset, and ERA5 reanalysis data, capturing the case of Tropical Cyclone CILIDA over the South Indian Ocean on December 24, 2018. The retrievals from both MWHS-I and MWHS-II effectively capture the spatial distribution of high-IWP regions within the cyclone's convective core, a feature that is also accurately characterized by the CCIC product. In contrast, while the ERA5 reanalysis dataset broadly reproduces the macroscopic structure of these high-IWP regions, it exhibits significantly lower spatial detail compared to the satellite retrieval products.

To further evaluate performance against the CCIC product and the narrow-swath 2C-ICE observations, we performed spatiotemporal collocation and generated scatter plots for quantitative analysis. As illustrated in the scatter plots, the retrievals from MWHS-II demonstrate a higher degree of agreement with both the CCIC and 2C-ICE benchmarks compared to MWHS-I. This indicates a substantial improvement in retrieval capability and performance for the second-generation instrument relative to its predecessor.

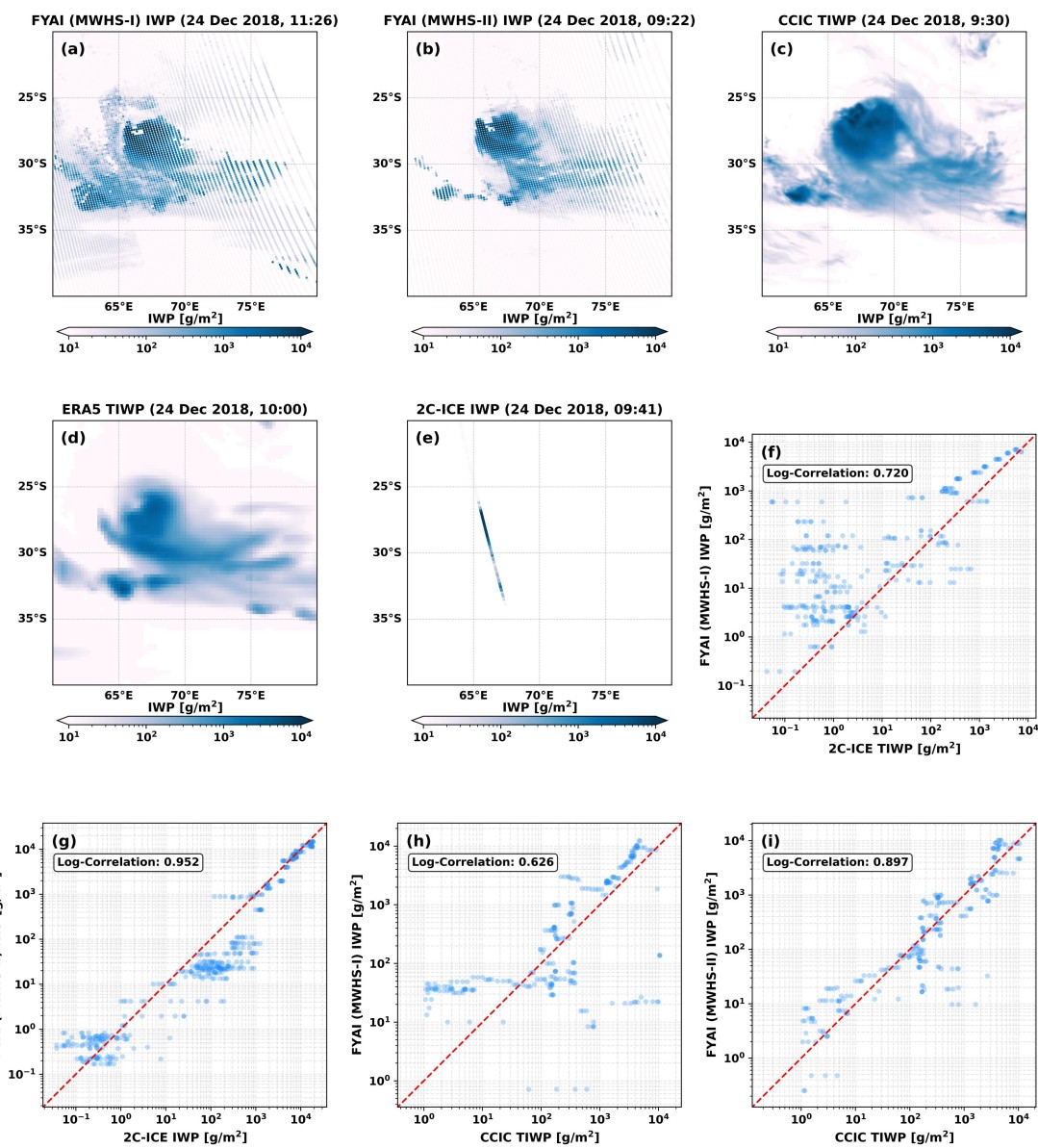

**Figure 6: Comparison of FYAI L2 IWPs from MWHS-I and MWHS-II retrieval, CCIC, 2C-ICE and ERA5 in a case study of tropical cyclone. UTC time is used.**

## 6.2 Global gridded product comparison and zonal mean comparison

Figure 7 presents the multiyear average spatial distribution of the IWP, whereas Figure 8 shows the zonal mean distribution of the IWP. All the IWP products were resampled to a spatial resolution of (1°×1°). All the IWP products exhibit fundamentally consistent spatial patterns. Notably, FYAI demonstrates closer alignment with active sensor products than passive ones. However, it is important to point out that compared to the 2C-ICE and DARDAR active remote sensing baselines, the IWP retrieved from MWHS-II shows a slight overestimation in the equatorial region. In contrast, the MWHS-I retrievals align more closely with active observations at these latitudes. Meanwhile, both

MWHS-I and MWHS-II exhibit a notable underestimation in the mid-to-high latitudes of the Southern Hemisphere. Although the time series do not overlap, we selected the 2007-2010 period for active instrument comparison because of CloudSat's superior data completeness before 2011. This selection is necessitated by data constraints but remains scientifically justified, as both spatial patterns and total magnitudes show minimal variation in IWP sequences. Additionally, passive optical/infrared instruments (MODIS, VIIRS) and the ERA5 reanalysis result in significant underestimations of IWP values at low-to-mid latitudes, whereas the MODIS and VIIRS retrieval products result in substantial overestimations in polar regions. For the SIWP, the multiyear average spatial distribution and zonal mean are shown in Figure 9 and Figure 10 ; the overall distribution closely resembles that of IWP, but the values are lower in magnitude. Notably, the SIWP derived from FYAI MWHS-II shows a closer agreement with 2C-ICE.

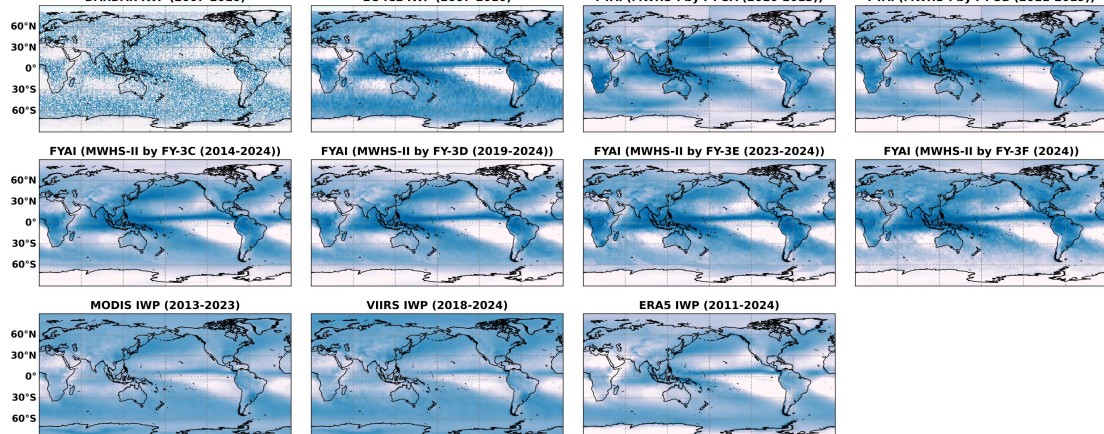

**Figure 7: Global average spatial distributions of the IWP compared with those of other satellite products and reanalysis products.**

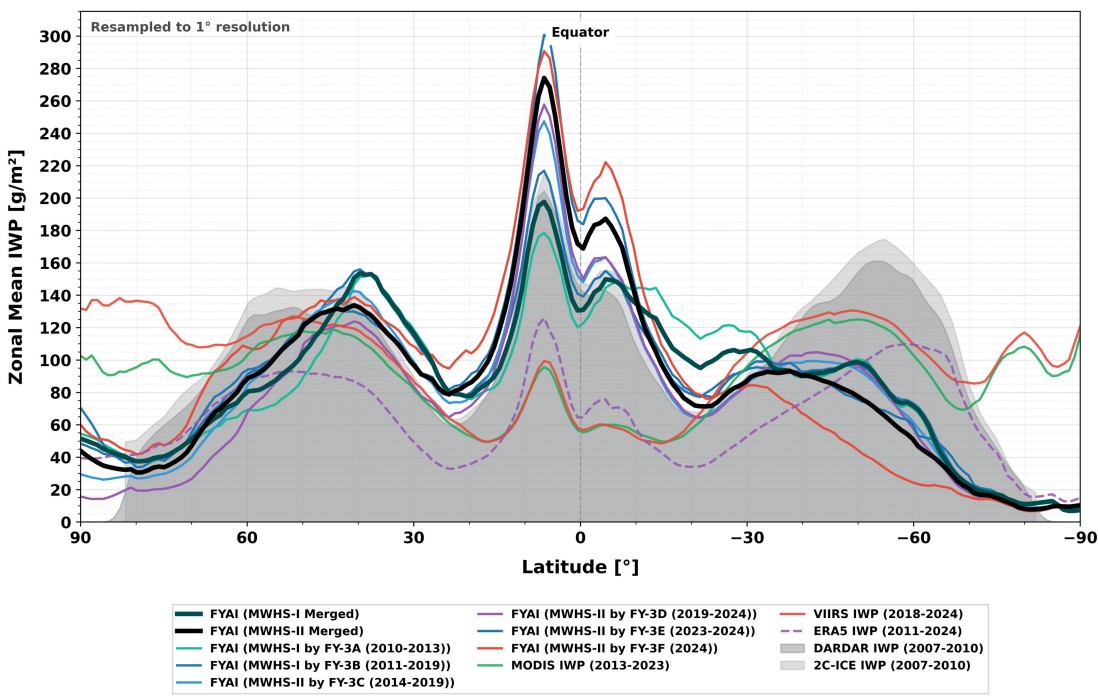

414

415  **Figure 8: Zonal mean IWP compared with other satellite products and the ERA5 reanalysis.**

416

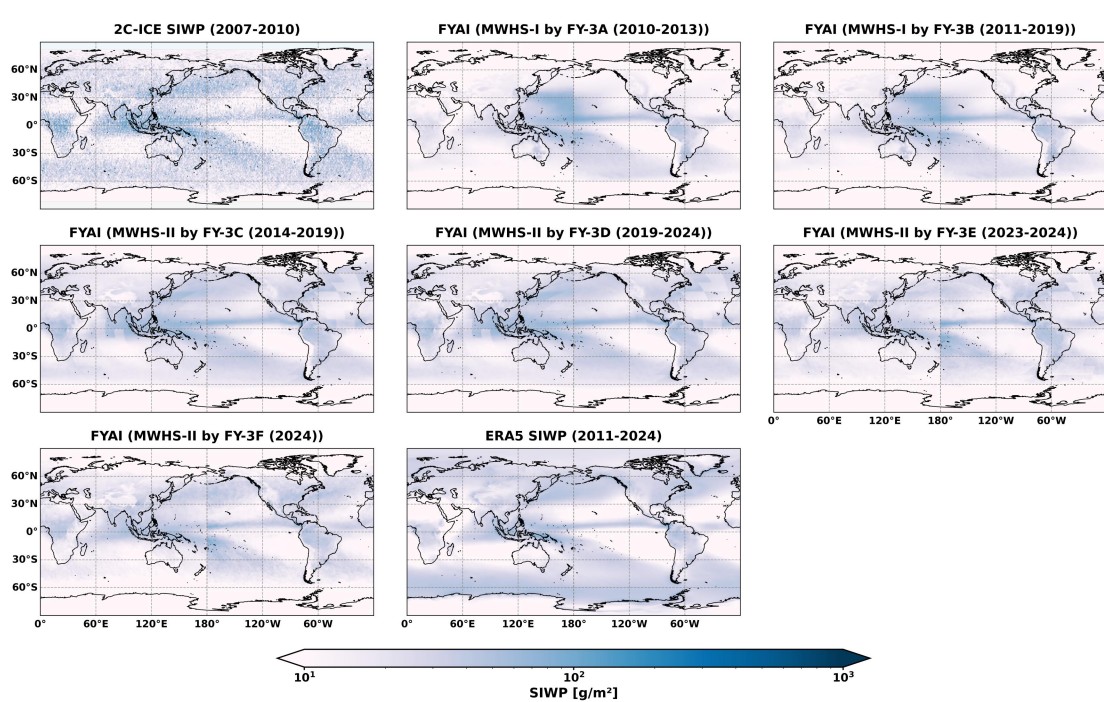

417

418  **Figure 9 : Analogous to Figure 7 but for SIWP.**

419

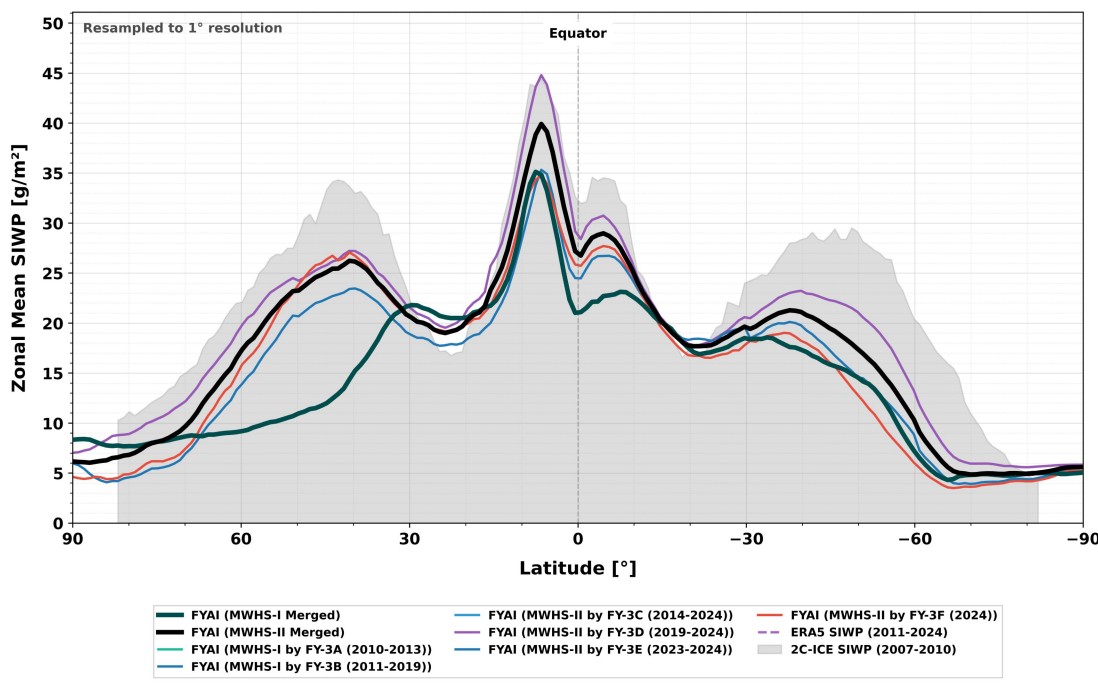

**Figure 10 : Analogous to Figure 8 but for SIWP.**

**6.3 Long-term analysis of gridded products**

Figure 11 presents the time series of global total atmospheric ice mass derived from our gridded retrieval products for the period of 2011 – 2024. For comparison, the orange and blue-green lines represent IWP data from 2C-ICE and DARDAR (another IWP product based on active remote sensing instruments; Delanoë and Hogan, 2008), respectively. Due to battery anomalies with CloudSat after 2011, which resulted in the loss of nighttime data, the time series for both 2C-ICE and DARDAR are restricted to the 2007–2010 period.

In terms of magnitude, our retrieval products align closely with 2C-ICE and DARDAR. In contrast, estimates from passive optical/infrared instruments (MODIS and VIIRS) and ERA5 reanalysis are significantly lower than the active radar-based baselines. Note that all mass calculations are area-weighted by latitude.

However, the time series reveals that the FYAI product exhibits larger interannual variability compared to the 2C-ICE baseline. This variability is not uniform over time; it is most pronounced during the FY-3B era. While variability decreases in the later period, the fluctuations in the early record likely reflect sensitivity differences inherent to the first-generation instrument. The mean global total

atmospheric ice mass from our products for 2011–2024 is 57.62 ± 2.32 Gt (calculated as the mean ±

one standard deviation based on a t-distribution; this also applies to the SIWP discussed below), which

is consistent with our previous estimation using the DARDAR product (Xu et al., 2022).

Regarding SIWP, retrievals from both MWHS-I and MWHS-II align closely with ERA5 and exhibit

strong consistency with the 2007–2010 2C-ICE baseline (Figure 12). The estimated global suspended

ice mass for the 2011–2024 period is 10.78 ± 0.99 Gt.

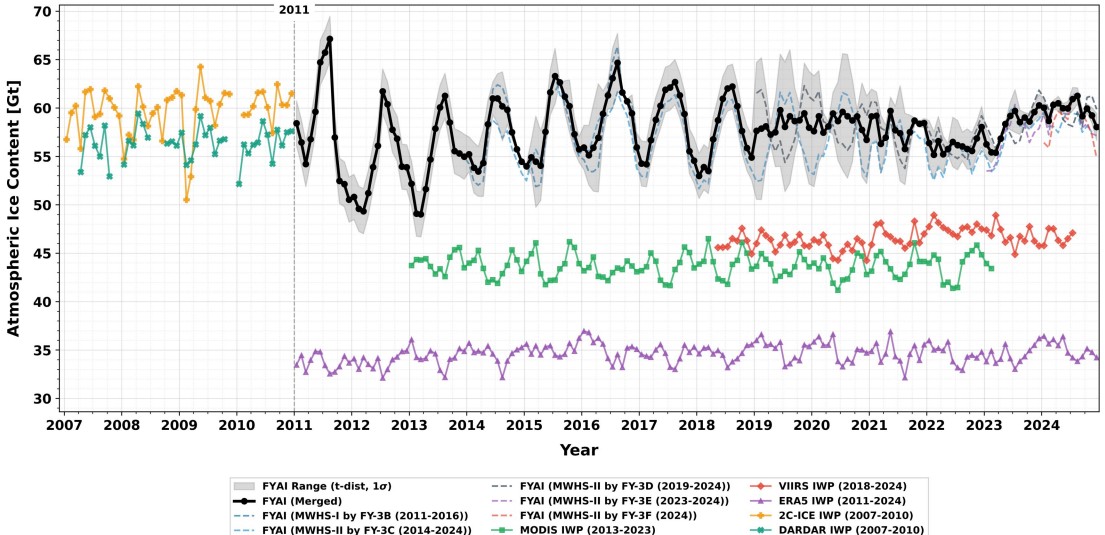

**Figure 11: Native time series of the monthly global average of total atmospheric ice and comparison with other satellite products, along with the ERA5 reanalysis. All calculations of total atmospheric ice consider latitude area weighting.**

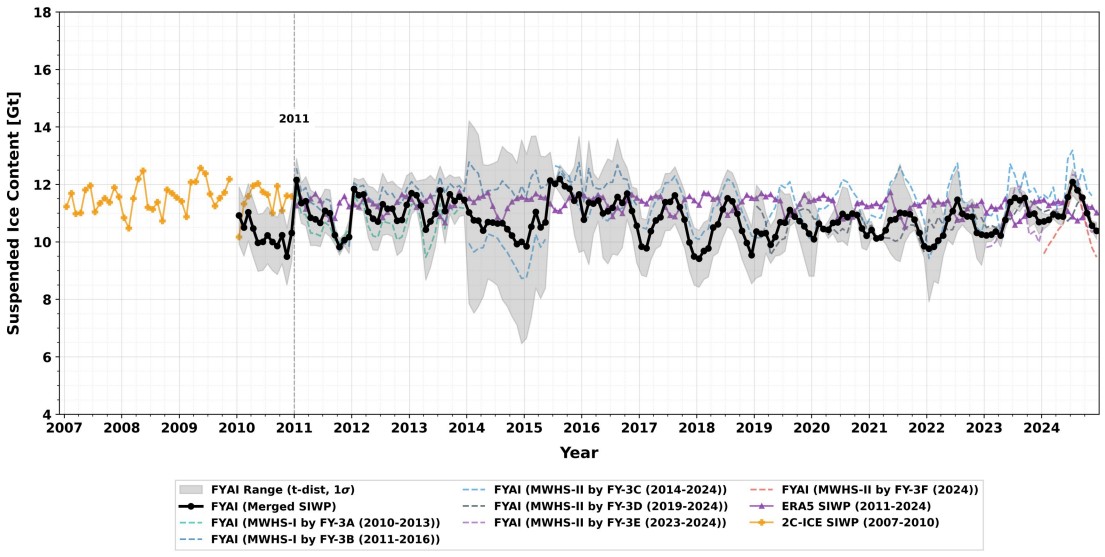

**Figure 12: Analogous to Figure 11. but for SIWP.**

453

**7 Uncertainty Analysis**

Although the uncertainty in IWC from 2C-ICE is approximately 30%, it remains one of the most reliable remote sensing IWP retrieval datasets currently available. As the FYAI dataset is generated using 2C-ICE as reference data for training machine learning models, it inevitably inherits uncertainty from 2C-ICE. This section outlines the uncertainty characterization for both FYAI L2 and L3 products.

**7.1 L2 Product Uncertainty**

The QRNN model employed in FYAI outputs an approximation of the quantile function (i.e., the inverse cumulative distribution function, or inverse CDF) of the conditional distribution. Consequently, the model implicitly models a conditional probability distribution, allowing for the retrieval of specific percentiles of the estimated variable. We have selected the 5th and 95th percentiles of the predicted distribution to represent the lower and upper bounds of uncertainty, respectively.

**7.2 L3 Product Uncertainty**

The uncertainty of the FYAI L3 product is calculated in two distinct stages. The first stage defines the uncertainty when aggregating L2 instantaneous observations into L3 monthly mean products, using the SEM as the metric. Based on the 5th/95th percentile bounds derived from the L2 products, and assuming errors follow a normal distribution, the variance for individual pixels is first estimated. Then, following the law of propagation of uncertainty (assuming independent errors among pixels within a grid cell), the variance of the grid mean is calculated (as the sum of individual variances divided by the square of the total number of observations falling within that grid). Finally, the square root of this variance is taken to obtain the monthly SEM.

The second stage addresses the uncertainty when aggregating L3 monthly means into L3 annual means. To avoid underestimating the final uncertainty, a conservative estimation strategy is adopted: assuming highly correlated errors between months (e.g., potential systematic errors), the annual mean uncertainty is defined simply as the arithmetic mean of the uncertainties of the 12 months in that year.

**8 Conclusion and usage notes**

A global IWP and SIWP dataset spanning 2010 – 2024 was produced using a machine-learning

framework derived from passive-microwave observations (MWHS-I/II) onboard the FY-3 satellite
series. Three distinct product levels were generated: (1) L2 IWP and SIWP preserving native sensor
resolution (15 km at nadir); and (2) L3 monthly gridded global composites (1° × 1°) for individual
sensors.

Prioritizing global representativeness and long-term homogeneity over instantaneous pixel-level
precision was a deliberate strategy in this study. While our passive microwave retrievals provide the
wide-swath coverage essential for decadal climate analysis, they may not match the instantaneous
accuracy of active sensors. We acknowledge that relying on 2C-ICE for training inevitably imparts the
reference product's systematic biases to our dataset. Furthermore, representativeness errors arise from
the spatial mismatch between the coarse MWHS footprint (~15 km) and the narrow 2C-ICE track.
Although the deep neural network effectively filters label noise by leveraging substantial data volumes
—capturing robust statistical relationships even under beam-filling constraints—it must be noted that
the reported error metrics likely underestimate the actual uncertainty in highly heterogeneous scenes.

Specific limitations regarding variable definition and instrument stability must be acknowledged. First,
the partition of SIWP from total IWP represents an exploratory effort. Since no single instrument
currently distinguishes suspended from falling ice reliably, this separation serves primarily to facilitate
model-observation comparisons. Second, regarding temporal stability, specific subsets of the FYAI
dataset require cautionary usage (summarized in Table 3 Summary of FYAI dataset components
requiring cautionary usage or having specific limitations). The larger interannual variability
observed in the FY-3B era reflects a necessary trade-off: lacking the 89 GHz channels available on
MWHS-II, we incorporated the 150 GHz channel to ensure sensitivity to ice clouds (Wang et al., 2022).
Unlike the opaque 183 GHz band, this window channel is susceptible to surface emissivity variations,
introducing background noise into the time series—a stability issue largely resolved in the post-2014
MWHS-II era. Additionally, L3 products derived from FY-3B show anomalous positive deviations
during 2017–2019, attributed to potential instrument aging. Conversely, FY-3A products (2010–2013)
exhibit a slight underestimation. While FY-3A and FY-3B form a valuable morning-afternoon
constellation, users should be aware of these calibration nuances when conducting long-term trend
analyses. We are actively working to address these issues in future updates through physics-based
constraints and close collaboration with instrument specialists.

Based on this methodology, we generated comprehensive retrieval products spanning FY-3A through
FY-3F. A distinctive advancement of this dataset is its global applicability over both land and ocean—
surpassing the ocean-only limitation of many existing passive microwave products.

**Table 3 Summary of FYAI dataset components requiring cautionary usage or having specific limitations**

| Satellite/Sensor Name | Time | Product Level | Note |
| --- | --- | --- | --- |
| FY-3A (MWHS-I) | 2010-2013 | L3 | Use with caution for long-term time series analysis. |
| FY-3B (MWHS-I) | 2017-2019 | L3 | Use with caution for long-term time series analysis. |
| FY-3C (MWHS-II) | 2015/5/31-2015/7/31 | L2, L3 | FY-3C operational service has been suspended since 31 May 2015 due to technical reasons. |


Looking ahead, we will explore advanced data fusion architectures to address current limitations. Our
future work will prioritize three key directions: (1) Synergetic retrievals combining passive microwave
with optical/infrared observations, utilizing cloud-top information to compensate for the microwave
spectrum's insensitivity to cirrus clouds; (2) Joint retrieval frameworks that simultaneously assimilate
multispectral observations within a unified radiative transfer model; and (3) Physics-Informed Neural
Networks (PINNs) that incorporate cloud microphysical constraints to enhance the accuracy of vertical
stratification.

In particular, the deployment of next-generation observation missions, such as EarthCARE and DQ-1,
will provide superior reference benchmarks. Integrating these high-fidelity datasets will allow us to
mitigate label noise and further refine retrieval accuracy. Furthermore, recognizing the rapid
advancements in terahertz remote sensing instrumentation (Li et al., 2023), we plan to leverage

terahertz technology to achieve higher-precision retrievals of IWP and SIWP. Collectively, these enhancements will significantly bolster the product's utility for monitoring rapidly evolving meteorological phenomena and validating climate model cloud parameterizations.

**9 Code and data availability**

The datasets generated in this study are available for download at https://doi.org/10.11888/Atmos.tpdc.303143 and https://cstr.cn/18406.11.Atmos.tpdc.303143, and should be cited as (Yang et al., 2025). Additionally, the code and model weights have been deposited at (Yang, 2025). Regarding the public source data used in this work, the FY-3 MWHS-I/II Level-1 observations are accessible via the National Satellite Meteorological Center (NSMC) data portal (https://data.nsmc.org.cn); the CloudSat-CALIPSO products (2C-ICE and 2B-CLDCLASS) can be obtained from the CloudSat Data Processing Center (https://data.nsmc.org.cn); the ERA5 reanalysis data are available via the Copernicus Climate Change Service (C3S) Climate Data Store (https://cds.climate.copernicus.eu/datasets/reanalysis-era5-single-levels?tab=overview) under the dataset "ERA5 hourly data on single levels from 1940 to present"; and the CCIC product is hosted on the Amazon Web Services (AWS) Open Data Registry (https://registry.opendata.aws/ccic/. AmazonWebSevicesOpenData).

**Author contributions.** YFY conceived the main algorithm, produced the dataset, validated its accuracy, and drafted the manuscript. GJX and RZ also contributed to parts of the algorithm design. BL, LTHS, WYW, CDX, and TFD supervised data production and validation, and revised the manuscript.

**Competing interests.** The contact author has declared that none of the authors has any competing interests.

**Acknowledgements.** The authors acknowledge the National Satellite Meteorological Center (NSMC) and CloudSat Data Processing Center for providing access to the satellite data utilized in this work.

**Financial support.** This research is supported by National Natural Science Foundation of China grant 42222608.

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
