# Peer review of "FYAI: A Fengyun Satellite-Based Dataset for"

_Earth System Science Data, 2025_

## Author Comment (AC1)

**Point-by-point Response to Reviewers**

**Manuscript ID: [essd-2025-447]**

**Title: FYAI: A Fengyun Satellite-Based Dataset for Atmospheric Ice Water Path**

**General Response**

We thank both reviewers, Prof. Patrick Eriksson and the anonymous reviewer, for their constructive comments, which have significantly strengthened the clarity and completeness of our manuscript and dataset.

In response to the overall feedback, we have revised the article title to "FYAI: A Fengyun Satellite-Based Dataset for Atmospheric Ice Water Path" to better align with the data-focused scope of ESSD. We have also substantially restructured the manuscript, shifting emphasis away from methodological details and toward a clearer description of the dataset itself and its availability.

Our detailed point-by-point responses follow this summary. Below are the highlights of our major revisions:

1. Release of Optimized Dataset: We have generated an optimized version of the dataset. It is currently being uploaded, and the specific DOI/URL will be provided in the revised manuscript to ensure seamless access.

2. Instrument Characteristics and Spatial Matching: We addressed the incomplete description of scanning geometry:

- Instrument Specifications: We now explicitly state that MWHS/MWHS-II are cross-track scanners and provide specific nadir resolutions.

- Resolution Mismatch: For the 89 GHz channel, we adopted a nearest-neighbor matching strategy.

- Beam Filling and Footprint: To address varying footprint sizes and Non-Uniform Beam Filling (NUBF), we added the Satellite Zenith Angle (SZA) as an input feature and applied a strict Coefficient of Variation (CV) filter.

3. Training Data Representativeness and Error Assessment We acknowledge the fundamental challenge of matching coarse MWHS footprints with narrow 2C-ICE tracks.

- Representativeness: While sparse sampling introduces label noise, we argue that the deep learning model is robust to this noise given the large dataset size.

- Error Limitations: We explicitly clarified that our error metrics likely underestimate uncertainty in highly heterogeneous scenes due to the lack of area-averaged ground truth.

4. Definition of CIWP/SIWP

- Clarification: We replaced "CIWP" with "Suspended Ice Water Path (SIWP)" to avoid ambiguity.

- Methodology: Extraction follows the FLAG methodology (Li et al., 2012), strictly excluding surface precipitation and deep convection.

- Motivation: SIWP provides a crucial observational constraint for the "cloud ice" variable in climate models.

5. Data Quality and Artifact Handling

- Artifact Removal: We identified that "LandCover" features caused striping artifacts and have retrained the model without them, effectively mitigating the issue.

- FY-3A Inclusion: Retrievals from FY-3A are now included in the full analysis, with appropriate caveats regarding its lower precision.

- Transparency: A new Table 3 summarizes periods with suboptimal quality.

6. Restructuring for ESSD (Data-Centric Focus)

- New "Data Records" Section: A dedicated section now comprehensively describes file formats, variables, resolution.

- CCIC Comparison: We introduced the Chalmers Cloud Ice Climatology (CCIC) as a primary benchmark for comparative analysis.

- Input/Output Tables: New tables explicitly list all input features and output variables.

We believe these revisions comprehensively address the reviewers' concerns and result in a manuscript that effectively presents the FYAI dataset. Below, we provide point-by-point responses to the reviewers' comments.

RC1:

The manuscript by Yang et al. presents a new dataset of retrievals based on the MWHS instrument series. These retrievals focus on the ice water path (IWP), but also cloud IWP (CIWP) and cloud mask are considered. Retrievals of IWP based on operational microwave radiometers are surprisingly few, despite some clear advantages of such measurements for the task. An important forerunner is the work of Holl et al. (2014), also applying machine learning, using the same reference dataset (2C-ICE) and making use of similar microwave radiometers. However, Holl et al. (2014) also included passive near and thermal infrared (IR) measurements and in such way increased the sensitivity at conditions matching lower IWP. On the other hand, by involving near-IR data a restriction to day-time was introduced, a limitation avoided in this work. Another strength of this work is the relatively long time series of data provided, in contrast to Holl et al. (2014) that so far not been applied in an operational manner.

That is, the retrievals presented fill an important gap, and we want to see this dataset description being published in ESSD. However, the manuscript requires a major revision; at least, the details of how these retrievals based on machine learning were developed must be better described and the characterization of the retrieval performance must be extended. Details behind this recommendation are elaborated below.

As there will be several references to our own work, including a suggestion to consider data produced by us, we've decided to not stay anonymous in the interest of transparency. This review is made by Patrick Eriksson, assisted by PhD student Peter McEvoy. That said, we think the references to our own work are motivated.

General comments:

- The description of input data is in some parts detailed, such as the quality filtering. The specifications of the instruments' channels are found in the Supplement (but the existence of the supplement is not mentioned). On the other hand, very basic information is missing. Most importantly, the scanning type of the instrument and the footprint sizes are ignored. According to the WMO OSCAR database, the MWHM instruments are cross-track scanners having a 183 GHz nadir footprint size of 16 km. The lower frequencies included in MWHS-2 have a 32 km nadir resolution. Two major concerns appear here, none discussed in the manuscript. For a cross-track scanner, the footprint sizes vary with position inside the swath. How is this handled in the training, and does the retrieval accuracy depend on scan angle? Further, for MWHS-2 the retrievals involve channels having different spatial resolutions. How is this handled?

    Response: We thank the reviewer for raising these important points regarding instrument

characteristics, and we acknowledge that our original description of the scanning geometry and spatial resolution of the MWHS/MWHS-II instruments was indeed incomplete.

We have updated the instrument description in Section [Input data] to address these points. The text now explicitly states that both MWHS and MWHS-II are cross-track scanners, and we provide the specific nadir resolutions for each channel.

For the 89 GHz channel resolution mismatch, we adopted a nearest-neighbor matching strategy. The brightness temperature from the larger 89 GHz footprint was directly assigned to the co-located finer-resolution (16 km) pixels without spatial interpolation. We believe the deep neural network can treat such feature-level spatial mismatches as stochastic noise during training.

To account for the varying footprint size across the scan swath, we included the Satellite Zenith Angle (SZA) as an input feature, allowing the model to learn scan-angle dependencies. We also applied a strict filter based on the Coefficient of Variation (CV) of CloudSat IWP to preferentially exclude highly heterogeneous scenes where footprint mismatch errors are most pronounced.

These clarifications have been incorporated into the revised manuscript.

- The footprint size is also of concern for the generation of the training database. With a MWHS resolution >= 16 km, even in the best case, only a fraction of the MWHS footprint is covered by the reference data (2C-ICE). The lack of footprint filling will result in erroneous training data, as the average IWP over the full footprint is not represented. This is an important concern, not discussed at all.

  Response: Thank you for raising this crucial point regarding the footprint size mismatch for training data generation. We fully acknowledge that the coarse spatial resolution of MWHS ($\geq$16 km) and the narrow along-track sampling of the 2C-ICE reference data pose a fundamental representativeness challenge. Since 2C-ICE covers only a small portion of an MWHS footprint, the reference IWP may not accurately represent the mean IWP across the entire field of view, especially in heterogeneous cloud scenes.

  In our collocation approach, we applied a fixed radial threshold (~7.5 km) relative to the MWHS footprint center, which corresponds roughly to the nominal semi-diameter at nadir. Although the MWHS resolution degrades toward the swath edge, we maintained a fixed threshold to ensure consistent sampling of the footprint core. Adopting a scan-angle-dependent threshold would primarily extend sampling along the one-dimensional CloudSat track rather than properly capturing the two-dimensional enlargement of the off-nadir footprint. Thus, the fundamental issue of sparse sampling remains.

We recognize that this spatial mismatch introduces representativeness noise into the training labels. However, deep learning models can be robust to such random label noise when trained on sufficiently large datasets. Because the misalignment between cloud structures and the satellite tracks is largely random, the model can still learn the bulk statistical relationship between brightness temperatures and footprint-averaged IWP. These "noisy" samples tend to average out during large-scale training.

In response, we have added a dedicated discussion in Section [Conclusion and usage notes] to explicitly acknowledge the beam-filling effect as a key source of uncertainty in the reference labels and to address its impact on retrieval performance, particularly in inhomogeneous scenes.

[Figure]

**Schematic diagram illustrating the variation of MWHS-II spatial resolution with scan angle.**

- The manuscript is not clear, but it seems that the same problem is present in the test data, resulting in that the true retrieval error can not be assessed, just the error with respect to a partial IWP over the footprint. There is also a concern that inhomogeneous situations (but still physically correct) situations are ignored, see further comment for line 145.

Response: Thank you for raising this critical issue regarding the representativeness of the training data due to the footprint size mismatch. We agree entirely that the coarse spatial resolution of the MWHS instruments ($\geq 16$ km at nadir) combined with the narrow along-track sampling of the 2C-ICE reference poses a fundamental beam-filling challenge. The reference IWP from the 2C-ICE track likely does not represent the true mean across the entire MWHS field of view, especially in heterogeneous cloud scenes.

In our collocation procedure, we used a fixed radial threshold (~7.5 km) centered on the MWHS footprint. This threshold approximates the instrument's nominal semi-diameter at nadir. Although the MWHS resolution degrades toward the swath edges, we maintained a fixed threshold to ensure consistent sampling of the footprint core. A geometrically precise, scan-angle-dependent threshold would primarily extend sampling along the one-dimensional CloudSat track, rather than adequately capturing the two-dimensional enlargement of the off-nadir footprint. Therefore, the underlying issue of sparse sampling remains even with a variable threshold.

Looking ahead, we plan to refine the collocation strategy in future work. A specific improvement will be to implement a scan-angle-dependent variable threshold, where the matching radius is explicitly calculated from the instrument's instantaneous field-of-view (IFOV) geometry. While this adjustment alone cannot overcome the fundamental sampling limitation of the 1D reference track, it will better align the matched data with the actual two-dimensional footprint at each scan position, improving the physical consistency of the training samples.

We recognize that this spatial mismatch introduces representativeness noise into the training labels. However, with a sufficiently large dataset, the deep learning model can be robust to such random label noise. Because the misalignment between cloud structures and the satellite tracks is largely random, the model can still learn the bulk statistical relationship between brightness temperatures and footprint-averaged IWP from the aggregate data.

Finally, regarding performance evaluation, we agree that our reported error metrics—based solely on pixels collocated with 2C-ICE—do not represent the algorithm's performance across the entire swath, particularly in heterogeneous scenes. This limitation is unavoidable, as no current observing system provides "true" area-averaged IWP ground truth for the large MWHS footprint. In heterogeneous conditions, the total error would be dominated by this representativeness error rather than the retrieval error of the algorithm itself. We have explicitly clarified in the revised manuscript section [Conclusion and usage notes] that our reported accuracy reflects performance under relatively homogeneous conditions and likely underestimates the uncertainty in highly variable scenes. We also discuss the beam-filling effect as a key source of uncertainty in the reference labels themselves.

- Besides IWP, cloud ice water path (CIWP) is retrieved. There exists no firm definition of cloud ice (see Eriksson et al. (2025) for our view on the topic), still the authors do not define what they consider as cloud ice. There is a reference to Li et al. (2012), but this is ambiguous information, as explained below (comments for line 157). However, cloud ice is normally taken to roughly match suspended ice hydrometeors, thus having a size below 150 um (Li et al., 2012). It has been shown that microwave observations at the frequencies of concern have low sensitivity to

such ice particles (e.g. Ekelund et al., 2020). Despite the low sensitivity, values can be extracted with ML, but the obtained results for CIWP should contain a low degree of direct measurement information; the CIWP is rather estimated through the correlation to IWP in the training dataset. Accordingly, the authors must motivate why the retrieval of CIWP is included. If it is kept, the limitations of this quantity must be analyzed and properly described.

Response: We thank the reviewer for the valuable comments and for directing us to (Eriksson et al., 2025), which provided deep insights into the definition and retrieval of cloud ice.

o   Motivation: Since climate models distinguish between suspended "cloud ice" and precipitating "snow", our objective is to retrieve Suspended IWP (SIWP) as an observational proxy for the "cloud ice" component, addressing the lack of constraints for model evaluation.

o   Physical Limitations: We agree that microwave observations (89–183 GHz) are insensitive to small ice particles (<150 μm). Therefore, our retrieval is indirect, relying on the statistical correlation between the strong scattering signals of precipitating ice and the co-existing suspended ice, as learned from the 2C-ICE dataset.

o   Revisions: In Section [Collocations], we have explicitly defined CIWP as suspended ice and clarified that this product represents an estimate based on microphysical correlations rather than direct detection.

- In fact, by not including any infrared data (as done by Holl et al. (2011)) the sensitivity to smaller ice crystals and lower IWP is limited. This is acknowledged on line 381, but is not properly discussed in the main text or explored in the error characterization. The latter is mainly done in such way that the errors for IWP below about 100 g/m$^2$ are difficult to discern. In particular, the performance when true IWP is zero is not clarified.

Response: We appreciate this critical observation. We acknowledge that without infrared data, sensitivity to small ice crystals and low IWP is inherently limited due to weak microwave scattering signals. In the revised manuscript, we have updated the evaluation to address this:

o   Performance at Zero IWP: We evaluated clear-sky performance using a 0.5 g/m² hard cutoff. We explicitly acknowledge that the False Alarm Rate (FAR) remains high (> 0.5). We explain in Section [Conclusion and usage notes] that this is physically rooted in the challenge of distinguishing the weak scattering signatures of tenuous clouds from radiometric noise and background fluctuations (e.g., surface emissivity or water vapor

variability).

- o Future Work: These limitations highlight the necessity of multi-sensor fusion—especially the integration of sub-millimeter wave data—which we have identified as a priority for future work.

- A CLM product is mentioned several times, including in the abstract, but no results from this product are shown. In fact, not even the nature of this product is described. Is this product holding cloud probabilities or binary mask (cloudy or not)? What vertical resolution?

  Response: In the optimized version of the dataset, we have removed the CLM product, as it is essentially an intermediate by-product of the data processing pipeline. We have also removed the corresponding references in the revised manuscript to avoid confusion.

- Several statistical measures of the retrieval performance are derived, but these all assumes that the reference dataset is error free. The errors inherited from 2C-ICE must somehow be represented and brought forward. The results are compared to several other datasets, but they are all (beside DARDAR) known to have considerable biases with respect to 2C-ICE (Eliasson et al. (2011); Duncan and Eriksson (2018)) and these comparisons bring little value. On this side we can not avoid bringing up the Chalmers Cloud Ice Climatology (CCIC, Amell et al. (2024)), having important overlap with this work (also machine learning using 2C-ICE as reference). For averaged values, CCIC matches 2C-ICE well, and as such should constitute a more interesting dataset for comparison. As CCIC is based on infrared measurements and in this work only microwaves are considered, there is value in contrasting the strengths and weaknesses of the two dataset. As CCIC is quasi-global with 30 min resolution, even local retrievals can be compared.

  Response: We appreciate the recommendation of Amell et al. (2024) and agree that comparisons with biased datasets add limited value. We have restructured the evaluation section to focus on:

  - o Inherited Errors: We now explicitly acknowledge that our retrieval assumes 2C-ICE as the "truth" and inherently inherits its uncertainties.
  - o Comparison with CCIC: We introduced the Chalmers Cloud Ice Climatology (CCIC) as the primary benchmark. This allows a valuable contrast between our LEO-based Microwave retrieval and the GEO-based Infrared CCIC.

- Bad retrievals are mentioned here and there, but it is very hard to get an overview of what data to avoid and how these have been handled. Are bad retrievals included on gridded monthly means, and how they have been handled within the manuscript (e.g. when producing Fig. 6)? A complication here is that measurements are referred to by both using the instrument and satellite names. Adding a table with periods and areas with bad retrievals would help. That said, all retrievals that the manuscript cover shall be considered as results, and e.g. FY3A shall be included in Fig. 7.

  Response: We genuinely appreciate this constructive comment. It has helped us significantly improve the consistency and clarity of our dataset and manuscript.

  - Model Optimization and Artifact Removal: We have refined the retrieval model. In the revised version, the artifacts previously observed have been effectively mitigated, and the retrieval accuracy of FY-3A has been improved to a usable level.
  - Inclusion of All Retrievals: Following your suggestion, we now treat all retrievals, including FY-3A, as valid results throughout the manuscript. FY-3A is now explicitly included in the analysis and figures.
  - Data Quality and Caveats: While we have included FY-3A, we maintain a transparent discussion regarding its quality. We explicitly note that the reliability of FY-3A retrievals is slightly lower than that of later instruments. We recommend users prioritize the other satellites for high-precision applications.
  - Dataset Naming (FYAI): To resolve the confusion between instrument and satellite names, we have formally named our dataset "Fengyun Atmospheric Ice" (FYAI). This allows us to refer to the products in a unified format.
  - Quality Table: As suggested, we have added a table (Table 3) summarizing the periods with lower retrieval quality or data gaps to guide users.

- The language is in many parts not clear. Examples are found below, but this shall not be taken as a complete list of language issues.

Specific comments

Line 21: According to the tables in the supplement, the instruments of concern can not be said to be "high-noise". It can also be questioned if noise is the main challenge in these inversions, this would be an ill-posed problem even in the limit of zero noise.

Response: The phrase *"high-noise satellite data"* has been revised to *"inherent uncertainties in satellite brightness temperatures and the spatial mismatch between passive microwave footprints and active radar/lidar training data."*

We acknowledge the reviewer's point that MWHS instruments are not *"high-noise"* in terms of

instrument NEDT (Noise Equivalent Delta Temperature). The primary *"noise"* in our context refers to the label noise or representativeness error arising from matching the narrow-beam CloudSat track (approx. 1.4 km) with the much larger MWHS footprint (>=15 km). This geometric mismatch, rather than sensor thermal noise, is the main challenge our QRNN model is designed to mitigate.

Line 23: It is unclear what is meant with "synoptic type that orbital-resolution", but presumably this refers to what is normally denoted to as level 2. The standard nomenclature of level 2 and 3 data should be adopted, see e.g. https://www.earthdata.nasa.gov/learn/earth-observation-data-basics/data-processing-levels

Response: We have replaced *"synoptic type"* and *"climatic type"* with the standard *"Level-2 (L2) products"* and *"Level-3 (L3) products"*, respectively.

Line 26: Where is there a compromise between the accuracy on footprint level and the spatial-temporal completeness?

Response: The phrasing has been adjusted to: *"FYAI bridges the gap between instantaneous pixel-level precision and broad spatiotemporal coverage."*

The "compromise" refers to the trade-off inherent in passive microwave (PMW) retrievals compared to active sensors (like CloudSat). While active sensors offer superior vertical profiling and pointwise accuracy, their narrow swaths limit global coverage and revisit times. PMW sensors (like MWHS) sacrifice some vertical resolution and pointwise precision (due to beam-filling effects) but gain wide-swath capabilities essential for capturing the full diurnal cycle and global spatial patterns.

Line 27: The statement of "unprecedented" is vague and can be questioned. With respect to understanding the cloud feedback, for example, the retrievals based on MODIS must be considered as equally or more interesting. In any case, the CCIC retrievals (Amell et al. (2024); Pfreundschuh et al. (2025)) have a much higher spatial-temporal coverage, still offering a similar accuracy (as also trained on 2C-ICE).

Response: The word *"unprecedented"* has been removed. The sentence now focuses on the dataset's specific utility: *"offering a comprehensive, decadal-scale record of global atmospheric ice content."*

We accept the critique. Claims of being *"unprecedented"* can be subjective. While MODIS provides high-resolution optical observations essential for cloud microphysics and CCIC offers exceptional temporal coverage from geostationary platforms, FYAI contributes a unique value proposition through its all-weather microwave capabilities and the inclusion of the dawn-dusk orbit from FY-3E. By filling

critical gaps in the diurnal cycle and offering a consistent, independent polar-orbiting perspective spanning 15 years, FYAI serves as a vital addition to the global IWP dataset family. It stands ready to function as a robust independent benchmark for validating climate models and cross-referencing other remote sensing records, thereby enriching our collective understanding of long-term atmospheric trends.

Lines 32-36: The impact of cloud ice on the radiation budget is brought forward as the main motivation, but as the measurements of concern do not constrain the amount of cloud ice in a direct manner (as discussed above), other passive observations are more relevant for this aspect. On the other hand, the relatively direct measurement of larger ice hydrometeors is of high relevance for e.g. distribution of latent heat and understanding precipitation processes. That is, the motivation to bring forward should be considered.

Response: Thank you for your insightful comment. We have revised the relevant statement to: *"Ice crystals play a pivotal role in cloud and precipitation processes, thereby significantly modulating the hydrological cycle, thermodynamics, and radiative transfer (Gultepe et al., 2017). Consequently, the reliable quantification of atmospheric ice content is critical for elucidating latent heat distribution and precipitation mechanisms (Amell et al., 2022)."*

We agree with your assessment. Since passive microwave observations are not primarily sensitive to the smaller cloud ice particles that dominate radiative forcing, framing the motivation solely around the radiative budget was not entirely appropriate. However, as you noted, atmospheric ice is highly relevant to precipitation processes and latent heat distribution, and it remains a significant source of uncertainty in climate models. Therefore, we have realigned the motivation to emphasize the hydrological and thermodynamic aspects, which better aligns with the capabilities of the observations used in this study.

Line 37: The statement of discrepancies in climate models "by orders of magnitude" needs closer specification. It is not true for mean IWP.

Resopnse: We have revised the relevant statement to: *"However, current climate models exhibit widespread inconsistencies and pronounced spatial heterogeneity in simulating IWP."* We believe this phrasing is more rigorous and accurately reflects the nature of model discrepancies.

Line 49: Much of our knowledge in this matter goes back to work by Frank Evans, e.g. Evans and Stephens (1995), and seems reasonable to cite any of those works (as done by Zhao and Weng (2002)).

Response: The manuscript has been updated to reflect this comment.

Line 51: "vertical profiles of the IWP"; IWP is a column value.

Response: Corrected as suggested.

Line 64: The logic in these two sentences is not clear. Rephrase.

Response:We have revised the narrative logic and updated the text to: *"However, the potential of China's Fengyun-3 (FY-3) series satellites remains largely untapped in producing global climate datasets. The FY-3 series offers a unique advantage unmatched by other operational systems: a complete three-orbit constellation comprising morning (FY-3A/C/F), afternoon (FY-3B/D), and the distinct dawn-dusk (FY-3E) orbit satellites (An et al., 2023; Tan et al., 2019; Wang et al., 2022). This configuration allows for substantially improved temporal sampling, filling critical gaps in the diurnal cycle of IWP that are missed by sun-synchronous satellites restricted to fixed crossing times, particularly with the inclusion of FY-3E observations starting in 2023. By leveraging this 15-year continuous archive (2010-2024), there is an opportunity to construct a coherent, long-term IWP climate data record that overcomes the spatiotemporal limitations of existing datasets."* We have shifted our focus to highlighting the current underutilization of the Fengyun series satellites in the development of global climate datasets, as well as emphasizing their distinct advantages.

Line 81: Please replace Amell (2021) with the related journal publication Amell et al. (2022).

Response: Corrected as suggested.

Line 83: The statement about Tana et al. (2025) does not seem correct. This was achieved, at least, in Amell et al. (2024).

Response: We appreciate your correction regarding the citation context. We have revised the statement to accurately reflect the contributions of previous studies, explicitly including Amell et al. (2024). The relevant text now reads: *"Previous efforts, such as SPARE-ICE* (Holl et al., 2014) *or geostationary retrievals (Amell et al., 2022, 2024; Tana et al., 2025), have demonstrated the efficacy of NN-based approaches ".*

Line 104: Wang et al. (2024) does not exist in the reference list.

Response: This reference has been added.

Line 105: Tables S1-S4 are referenced in the text. It is very unclear that this refers to tables within the supplemental material. Please clarify that there is a supplement.

Response: We have clarified in the text that these tables are provided in the Supplementary Material.

Line 105-106: The meaning of this sentence is unclear. What else than L1 should be used as basis for the retrievals?

Response: We appreciate your comment and acknowledge that the original description was ambiguous. We wish to clarify that all input data utilized in this study are exclusively derived from the official Fengyun-3 L1 products. We have revised the relevant statement in the manuscript to ensure clarity and remove any potential confusion.

Line 124-135: The quality control of the input Fengyun data is presented. Was any quality control or filtering applied to the 2C-ICE reference data?

Response: We appreciate your question regarding the quality control of the reference data. In the revised manuscript, we have explicitly described the filtering process applied to the 2C-ICE product. The relevant text has been updated to state: *"Additionally, for the 2C-ICE product, we excluded data points where the 'Data_quality' variable was non-zero, as a value of 0 indicates good data quality."*

Line 140: FY-3D and CloudSat are presented to be 30 min apart. If correct, there should not be any tropical collocations inside 15 min.

Response: We appreciate your scrutiny regarding the orbital timing. While the nominal Local Time of Ascending Node (LTAN) suggests a 30-minute separation (14:00 for FY-3D vs. 13:30 for CloudSat), the actual orbital dynamics are more complex. CloudSat has undergone multiple orbital maneuvers throughout its mission life, causing its LTAN to drift from the strict 13:30 schedule. Similarly, FY-3D is designed to allow for an LTAN drift of up to 15 minutes over a two-year period. Consequently, the relative positions of the two satellites do vary, and it is indeed possible for their overpass times to align within the 15-minute window. Our collocation process empirically confirmed the existence of these matches.

Lines 144: "pixel" seems to here mean boresight, but pixel indicates an area and is easily interpreted as footprint.

Response: We appreciate the reviewer for pointing out this terminology ambiguity. We agree that "pixel" can be misleading when referring to the specific geolocation of an observation. In the revised manuscript, we have replaced "pixel" with "field of view (FOV)" or "footprint" to denote the observation area, and "FOV center" to denote the precise geolocation (boresight), ensuring clearer distinction between the area and its center coordinates.

Line 145: This second criterion states, limiting co-locations to cases where the coefficient of variation for 2C-ICE pixels within an MWHS-II pixel is less than 0.6. This introduces a bias due to training only on relatively uniform cases. It would be helpful to have more motivation for this choice. Further, it must be clarified how the removed cases are considered in the error characterization.

Response: We appreciate the reviewer's critical observation regarding the data filtering criterion (CV < 0.6). We have addressed this concern from three aspects in the revised manuscript:

1. Motivation (Mitigating Representativeness Errors): The primary motivation for this threshold is to mitigate the geometric mismatch and beam-filling effects inherent in collocating wide-swath passive microwave observations (≥15 km) with narrow-track active radar/lidar profiles (~1.4 km). In highly heterogeneous scenes (high CV), the spatially averaged IWP derived from the narrow 2C-ICE track often fails to represent the bulk ice mass within the much larger MWHS footprint that determines the observed brightness temperature depression. Including these cases would introduce substantial label noise, distorting the physical mapping between radiometric signals and IWP during training. The threshold of 0.6 is selected based on established practices in previous studies (Holl et al., 2010; Wang et al., 2022) to balance data volume and label quality.

2. Training Strategy: By filtering for relatively uniform scenes, we ensure the neural network learns the robust, fundamental physical relationships between brightness temperature depressions and ice scattering, rather than overfitting to geometric artifacts.

3. Error Characterization and Clarification: We acknowledge that this filtering introduces a selection bias, meaning our reported validation metrics primarily reflect the model's performance in relatively homogeneous cloud regimes. To clarify how the excluded cases are considered, we have explicitly added a cautionary note in the section [Conclusion and Usage Notes] of the revised manuscript: *"It must be noted that the reported error metrics likely underestimate the actual uncertainty in highly heterogeneous scenes."* This ensures transparency regarding the model's limitations in complex, broken cloud scenarios.

Line 149: Please clarify what is meant by uniform distribution across latitude bands and how

that is achieved.

Response: In the revised manuscript, we have removed this sampling strategy.

Previously, we attempted to downsample the data to achieve a uniform distribution because polar-orbiting satellites naturally provide a much higher density of observations at high latitudes. However, in the updated model, we opted to utilize all valid collocated points without filtering. We believe that maximizing the training data volume is more beneficial than enforcing a uniform latitudinal distribution, even though this results in a larger number of samples in polar regions.

Lines 149-153: Please clarify if and how these multiple training subsets are used. Or are they combined in some manner? Is there a separate model trained for each combination of MWHS-II and MWHS-I with IWP and CIWP?

Response: To clarify, we trained a total of four distinct models, corresponding to each specific combination of the two sensors and the two retrieval targets:

1. MWHS-I retrieving IWP

2. MWHS-I retrieving SIWP (referred to as CIWP in the previous version)

3. MWHS-II retrieving IWP

4. MWHS-II retrieving SIWP

We have updated the manuscript to explicitly state that independent models were trained for each of these four configurations.

Line 155: What is a balanced representation? In any case, motivate why going away from using the actual statistics of the reference dataset.

Response: We appreciate your comment. In the updated model, we have abandoned the "balanced representation" approach and instead utilized the actual statistics from the reference dataset.

Line 157: As mentioned, just a reference to Li et al. (2012) is not sufficient. There is also a dot after (2012).

Response: We appreciate your suggestion. We agree that a simple reference was insufficient to fully explain the methodology. In the revised manuscript, we have expanded the description to explicitly detail how the suspended component was isolated. The updated text reads:

*"The calibration process for the SIWP training dataset followed an approach similar to that used for*

*the IWP dataset. Based on the FLAG methodology described by Li et al. (2012), we isolated the suspended component of the ice water path. This involved applying strict filtering criteria: all retrievals identified as surface precipitation were discarded. Furthermore, to minimize convective influence, we excluded data points classified as 'deep convection' or 'cumulus' according to the 2B-CLDCLASS product."*

Additionally, we have removed the extraneous period after the citation as noted.

Line 158: Why are the number of cases for CIWP and IWP not the same (there should exist CIWP value for IWP)?

Response: You are correct. In the previous version of the dataset, zero-value points were automatically filtered out from the CIWP dataset, resulting in a smaller sample size compared to IWP. However, in the revised manuscript and the updated QRNN model, we have rectified this inconsistency. The SIWP (formerly referred to as CIWP) and IWP datasets now share the exact same number of matched cases.

Line 159: As Sec 3 is very short (too short) it seems reasonable to merge Secs. 2 and 3.

Response: We appreciate this suggestion regarding the manuscript structure. We have reorganized and restructured the article to improve flow and coherence. The detailed outline of these structural changes is provided in the "General Response" section at the beginning of this document.

Line 161: How has this resolution been determined? It sounds unlikely as not all channels used have a resolution of 15 km, and this resolution is only reached at nadir.

Response: We appreciate your correction. We acknowledge that the resolution varies by channel and degrades away from the nadir. To avoid ambiguity, we have revised the description in the updated manuscript to explicitly state the resolution as *"nominal 15 km at nadir."*

Line 162-163: Please provide more details on how the monthly means are provided. Any weighting of the data? Are all grid cells filled? Typical number of retrievals in each mean? Are those numbers reported in the resulting data files? See also first data comment.

Response: We sincerely appreciate your valuable suggestion. In the optimized dataset, we have incorporated this information (such as sample counts). Furthermore, we have updated Table 2 in the revised manuscript to provide a detailed description of each variable, explicitly specifying its name, dimensions, and physical meaning.

Line 167: What is meant by "fundamental model" here? We can not find any RobustResMLP model outside this work. Or are the authors claiming to introduce RobustResMLP (but comments below contradict this)?

Response: We apologize for the confusion caused by our imprecise terminology."RobustResMLP" was the specific nomenclature we assigned to the MLP-backbone architecture developed in our previous iteration, and the term "fundamental model" was intended to refer to this underlying MLP structure. In the revised manuscript, we have removed these ambiguous expressions from the description of the updated QRNN model to ensure clarity and prevent any potential misunderstanding.

Line 169: Can 9 million parameters be considered as lightweight considering the few input data and the relatively limited scope of the model? For comparison, the MLP in Amell et al. (2022) had 0.3 million parameters. In any case, 9 million parameters seems large when compared to the training set of 700 000 – 900 000 cases. There should be a high risk for overfitting.

Response: We entirely agree with your assessment. Upon reflection, we recognized that our previous model architecture was unnecessarily complex (e.g., employing global attention mechanisms with $O(n^2)$ complexity), which indeed introduced a high risk of overfitting.

In the revised manuscript, adhering to the principle of parsimony (Occam's Razor), we have streamlined the architecture significantly. The updated QRNN model is a truly lightweight model with approximately 300,000 parameters, which aligns with the complexity levels observed in similar studies (e.g., Amell et al., 2022) and effectively mitigates the risk of overfitting given our training data size.

Line 170: "We make several significant improvements to the RobustResMLP" indicates that an existing model was used, but there is no reference to it.

Response: We apologize once again for the confusion caused by our unclear phrasing. Our original intention was to describe how we enhanced a standard MLP framework by integrating advanced techniques, such as attention mechanisms, to improve the model's predictive capability. To avoid any ambiguity regarding the model's origin, we have removed this specific phrasing from the description of the new QRNN model in the revised manuscript.

Line 170: This list is appreciated. However, for a reader in the geoscience community, these techniques may not be familiar. It would be very helpful to have references to articles or other

resources that provide more details on the techniques behind these improvements. Similarly, in Figure 1, references for "Lightweight Attention" and "Adaptive Feature Scaling" would be appreciated.

Response: We appreciate this helpful suggestion. We understand that specific deep learning architectures may not be familiar to all readers in the geoscience community. To address this, we have made the following revisions:

1. We have added relevant references for the specific technical modules (e.g., lightweight attention) in the main text to provide necessary context (He et al., 2016; Vaswani et al., 2017).

2. We have included the detailed hyperparameters of the new QRNN model in Table S5 of the Supplementary Material.

3. Furthermore, to provide a deeper understanding of our key architectural innovations—specifically the "Feature Gating and Bottleneck Attention Mechanism"—we have added a dedicated section (Text S1) in the Supplementary Material containing a detailed technical description.

Sec 4.1: Since this is a supervised machine learning, the retrievals will work as long as the scenario being observed is similar to those in the training dataset. How does it handle rare events that are not close to the training set? Is there a way for the method to identify or flag retrievals that risk being out-of-distribution?

Response: We appreciate the reviewer's insightful comment regarding the limitations of supervised learning in handling out-of-distribution (OOD) or rare events. To address this specific challenge, we have fundamentally upgraded our retrieval algorithm to a QRNN framework in the revised manuscript.

Unlike traditional regression models that output a deterministic point estimate (which provides no information about the retrieval confidence), the QRNN framework is designed to estimate the full conditional distribution of the target variable. This probabilistic approach provides a robust mechanism for identifying and flagging retrievals with potential risks:

1. Handling Rare Events: By estimating multiple conditional quantiles simultaneously, the model provides a comprehensive probabilistic view rather than a single value. When the model encounters rare events or complex scenarios that deviate from the training distribution, the predicted probability distribution typically widens.

2. Flagging High-Risk Retrievals: We explicitly utilize the 5th and 95th percentiles to define the uncertainty bounds for each retrieval. The width of this confidence interval serves as a dynamic indicator of retrieval reliability. A wide interval effectively "flags" instances where the model has high uncertainty—often corresponding to rare, noisy, or out-of-distribution

inputs—thereby allowing users to identify and filter these high-risk retrievals.

Furthermore, we have incorporated a deep residual architecture with attention mechanisms, which enhances the model's ability to focus on the most stable and critical feature channels, further improving robustness against input variability.

Line 186: Interesting solution on ensuring continuity across satellite generations by remapping values to the 150 GHz channel. However, at least a sentence or two quantifying any errors introduced by this approach, or providing motivation for why this can be expected to work sufficiently well, is motivated.

Response: In the revised model, we have excluded the 150 GHz and 166 GHz channels from the MWHS-II input. This decision was driven by two primary factors: first, as window channels, they are sensitive to surface emissivity, which can introduce background noise; second, our analysis indicated that their inclusion introduces additional errors. Therefore, we opted to remove these channels entirely, rendering the previously mentioned remapping strategy unnecessary.

Line 193: Should be Fig. 1.

Response: Corrected as suggested.

Figure 1: The first text box indicates that MWHS-I and II are used together. Presumably "and" should be "or".

Response: We appreciate your observation. We have corrected the text in the new Fig. 1 accordingly.

Figure 1: The second text box explains that auxiliary data are used, but this is not mentioned in the text.

Response: We appreciate your observation. To address this, we have added a new table (Table 1) in the revised manuscript that explicitly lists all input variables fed into the model, including the specific auxiliary data used.

Figure 1: The text box "MLP Based Model for Mapping 150 GHz to 166 GHz" contradicts what was written in line 186, where the MLP is described as mapping to 150 GHz.

Response: We appreciate your attention to detail in identifying this inconsistency. However, this issue

has been resolved in the revised manuscript. In the updated QRNN model for MWHS-II, we have excluded the 150 GHz and 166 GHz window channels from the input feature set. Consequently, the mapping step is no longer required and has been removed from the workflow.

Line 200: With a log-transform, it must be described how IWP=0 was handled.

Response: Zero values were replaced with a small positive value of $10^{-6}$ to facilitate the log-transformation.

Line 219: It must be clarified how "detect clouds" is defined, for both retrievals. In the case of MWHS a log-transform is used and then no retrieval will be strictly zero.

Response: We appreciate your comment. Previously, we employed a threshold-based method to define cloud presence (values below the threshold were labeled as "cloud-free"). However, to streamline the dataset and avoid ambiguity regarding the log-transformation, we have removed the separate Cloud Mask (CLM) product from the optimized dataset.

Line 229-230: The naming convention suggests the orbital retrieval data is level 1, even though it is level 2. Recommend removing or changing L1 to L2 in the filenames.

Response: We accept your suggestion. We have corrected the file naming convention in the optimized dataset to strictly adhere to the Level-2 (L2) nomenclature standards.

Line 237: The naming convention proposes that the gridded data is level 1, even though it is level 3. Recommend removing or changing L1 to L3 in the filenames.

Response: We accept your correction. Similarly, we have updated the filenames for the gridded products in the optimized dataset to correctly reflect the Level-3 (L3) nomenclature.

Line 240: Detail how "the most temporally stable" were identified and extracted.

Response: In the optimized dataset, we have adopted a more transparent approach. Instead of selectively filtering for "stable" periods, we now include the full retrieval record for all products. To ensure users are informed about data quality, we have explicitly identified periods with suboptimal retrieval performance and listed them as cautionary notes in Table 3 of the revised manuscript.

Line 241: We were unable to find "Merged_Global_Mean.nc" in the data portal.

Response: The optimized dataset is currently being uploaded. We will provide the updated data access link (DOI/URL) in the revised manuscript to ensure that the merged files and all other data products are fully accessible to reviewers and future users.

Sec 6: In brief, this section must be extended. The presented results must be discussed more carefully. For example, Fig. 3 having six panels of results is just commented in very brief terms. In addition, errors not covered by the present analysis must be incorporated, as indicated above.

Response: We appreciate your valuable feedback regarding the depth of the results discussion. In the revised manuscript, we have substantially rewritten this section to provide a more thorough, objective, and rigorous discussion of the generated dataset. We have structured this revised section to systematically evaluate the retrieval performance from multiple perspectives: We have structured this section to systematically evaluate the retrieval performance. First, we explicitly discussed the systematic biases inherited from the 2C-ICE ground truth. Second, we compared the quantitative metrics (R, RMSE) to highlight the advantage of MWHS-II channels. Third, we introduced Probability Density Functions (PDFs) alongside Q-Q plots on an independent test set to prove the model captures the full dynamic range and climatological statistics. Finally, we utilized classification metrics (FAR, CSI) to scrutinize the model's sensitivity and limitations specifically in identifying weak signals (low-IWP).

Sec 6: For what data are the statistics derived? Training or validation data should not be used here. It must be clarified that the test data are sampled in an unbiased way. They should represent a fully random selection.

Response: The statistics are derived from an independent test set. After updating the model, we re-conducted the performance evaluation, and the relevant details regarding the test data have been provided in the revised manuscript.

Sec 6: Since the model is trained and tested on 2C-ICE data, any bias and error from this dataset will be inherited. This issue should be discussed and the magnitude of these inherited errors should be listed.

Response: We appreciate your suggestion. In the revised manuscript, we have added a discussion in the [Uncertainty Analysis] section regarding the biases of the 2C-ICE product, specifically regarding its deviation from in-situ observations. The text now states: *"Previous research indicates that the*

*uncertainty in IWC from 2C-ICE is approximately 30-40%, establishing it as one of the most reliable remote sensing IWP retrieval datasets currently available (Deng et al., 2013; Eriksson et al., 2025). As the FYAI dataset is generated using 2C-ICE as reference data for training machine learning models, it inevitably inherits uncertainty from 2C-ICE."*

Sec 6.1: We suggest making a figure with the occurrence fraction (histogram of values) of IWP in the training and test data, and retrieved dataset. This would clarify the nature of the training and test data, and also show the dynamic range of the retrievals.

Response: We have implemented this suggestion. We have created a new figure (Fig. 5) in the revised manuscript that illustrates the probability density functions (PDFs) of IWP for the training set, the test set, and the retrieval results on the test set. This figure clearly demonstrates the consistency of the data distributions and the dynamic range of the retrievals.

Line 253: Please quantify "high accuracy". The statement can be questioned as the biases reported are considerable.

Response: We agree with your assessment that the term "high accuracy" was subjective. We have removed this statement. While the model exhibits robust performance when $IWP > 10 \ g/m^2$, we acknowledge that larger biases exist in low-IWP regimes.

Line 256: Start a new paragraph when starting to discuss Table 3. Same at line 261, when moving to SHAP.

Response: Corrected as suggested. We have adjusted the paragraph structure to ensure a logical flow. Furthermore, as noted in our response to Line 261, the section regarding SHAP analysis has been removed to streamline the manuscript, which naturally resolves the transition issue at that point.

Lines 256-257: The statement referring to Table 3 must be explained, what results show this? The next sentence "Our analysis ...", is this an explanation to the previous sentence, or a new topic?

Response: We appreciate the opportunity to clarify this point. Our intention was to explain that for the MWHS-II model, increasing the number of input channels does not necessarily correlate with performance gains. Specifically, we observed that the brightness temperature distributions of the 118 GHz channels exhibit significant modal shifts during certain periods. Since our training dataset does not fully cover the temporal extent of these shifts, including these unstable channels introduced noise.

Consequently, we excluded them from the input feature set, which resulted in minimal performance loss on the test set while improving temporal stability. The sentence *"Our analysis..."* was intended to explain this selection logic. We have revised the text to make this explanation clearer.

Line 261: Clarify that Figs. S1 and S2 are in the supplementary material. What is SHAP?

Response: SHAP (SHapley Additive exPlanations) is a game-theoretic approach used to explain the output of machine learning models. It quantifies the importance of each feature by calculating its "marginal contribution" (i.e., Shapley value) to the model's prediction.

However, in the revised manuscript, we have shifted the primary focus to the presentation and characterization of the FYAI dataset itself. Consequently, to streamline the paper, we have removed the section regarding SHAP analysis. For further details on the SHAP methodology, we refer interested readers to Lundberg and Lee (2017).

Line 264: Wang et al. (2024) does not exist in reference list. In what way is there a consistency with Wang et al.?

Response: The corresponding reference has been included in the updated bibliography. Regarding the consistency with Wang et al. (2024), their study demonstrated that the inclusion of the 89 GHz channel yields significant improvements in retrieval performance. Our initial SHAP analysis also confirmed this finding. However, we have removed the SHAP analysis section in the revised manuscript to prioritize the description and characterization of the dataset itself.

Tables 2 and 3: Negative biases far exceeding the stated global mean IWP are reported. How is this possible. Are the retrievals giving negative values?

Response: We verify that our retrievals are strictly non-negative, as the model outputs are generated in logarithmic space and transformed back to linear space (or constrained by activation functions). The anomalous negative bias reported in the tables was unfortunately due to a calculation error in the statistical post-processing script used for generating the table. We have re-calculated all validation metrics to ensure their accuracy and physical consistency.

Table 3: Which combination of these inputs is the one applied for the general processing? Consider stating that clearly in the text, for example in Sec. 4. The two last combinations use lat and lon as input. Is that a wise choice? The training can then basically learn the geographical distribution (especially when using a net with millions of parameters). This

could maybe be OK for some application, but should be avoided if temporal changes and trends are considered. If the model applied generally takes lat and lon as input, the consequence of this choice must be explored.

Response: We thank the reviewer for this insightful comment. We agree that incorporating spatial coordinates (latitude and longitude) into deep learning models requires careful consideration to prevent the model from merely memorizing climatological spatial distributions at the expense of physical retrieval principles.

1. Justification for Using Lat/Lon: The inclusion of latitude and longitude is consistent with numerous established satellite retrieval algorithms in the literature (Gerrit Holl et al., 2014; Wang et al., 2022). The rationale is not to force the model to learn a static geographic map of IWP, but to use these coordinates as proxies for background geophysical parameters that are challenging to measure directly yet exhibit strong geographical variation. Specifically:

   o Atmospheric Profiles: The relationship between microwave brightness temperatures (BTs) and IWP is highly dependent on the atmospheric state (e.g., temperature lapse rate, water vapor profiles), which demonstrates significant latitudinal dependence (e.g., contrasting tropical and polar atmospheres).

   o Surface Background: Surface emissivity varies considerably with location (e.g., over deserts, vegetated land, and ocean). Given that MWHS channels (especially window channels) are influenced by surface emission, latitude and longitude help the model implicitly identify the surface background context, thereby improving its ability to isolate the ice cloud scattering signal.

2. Addressing Concerns about Trends/Time-Variation: We acknowledge the reviewer's valid concern regarding potential over-reliance on location. In our model architecture, however, the lat/lon features serve more as constraints (or priors) rather than the primary drivers. The dynamic variation in the output IWP remains predominantly driven by real-time changes in the input brightness temperatures (BTs). The neural network learns to interpret the same BT vector differently depending on whether it was observed in the tropics or at high latitudes—a physically sound approach due to the differing atmospheric backgrounds in these regions.

Figure 3: As IWP spans order of magnitudes, panels c and d should have a logarithmic x-axis, and the y-axis in f should give the relative error. As pointed out above, the errors when true IWP is zero must be reported somehow. For the first bar in panel b, please note that any retrieved IWP > 0 for (true) IWP=0 corresponds to an infinite relative error.

Response: We have revised the figure (now Fig. 4) to incorporate your suggestions, including the use of logarithmic axes to better represent the dynamic range.

Regarding the performance at True IWP = 0, we acknowledge that the relative error becomes mathematically undefined (infinite) and that retrieving zero-values is inherently challenging. We openly acknowledge that the model's performance in strictly zero-IWP scenarios is limited. To rigorously evaluate performance in this low-end regime without encountering numerical singularities, we adopted a threshold-based approach (using 0.5 g/m² as a divider). This allows us to assess the model's detection capability for weak signals versus clear-sky conditions using classification metrics (such as False Alarm Rate), rather than relying solely on regression errors.

Sec. 7: The section only deals with IWP. There are no validations or comparisons for the other variables: CIWP and cloud mask. Such validations should also be performed.

Response: We appreciate this feedback. We have restructured Section [Product validation] to include a comprehensive validation of SIWP (formerly CIWP) alongside IWP, ensuring a thorough assessment of all retrieved products.

Sec. 7: The section seems to ignore FY-3A. Does that indicate that those retrievals not are trustwhorthy? Include FY-3A, or remove it from the article (and disseminated data).

Response: We have included the FY-3A retrieval results in the revised section to ensure the dataset's completeness. It is important to note that the retrieval accuracy for FY-3A is generally lower than that of subsequent missions, likely due to the limitations and aging of the first-generation instrument. We are currently collaborating with instrument specialists to investigate and mitigate these calibration issues. In the meantime, we have transparently reported these discrepancies and added specific cautionary notes for FY-3A data usage in Table 3.

Sec. 7: Consider to include CCIC, as this is a product developed with similar objectives.

Response: We appreciate this valuable suggestion. We agree that the Chalmers Cloud Ice Climatology (CCIC) serves as a critical benchmark given its similar retrieval objectives.

1.  Inclusion of CCIC: In the revised manuscript, we have explicitly included CCIC in our comparative analysis (see Section [Typhoon events] and Fig. 6).

2.  Case Study Validation: Our analysis reveals a strong agreement between FYAI and CCIC, particularly in the case study of a tropical cyclone passage. This consistency validates the capability of FYAI in capturing intense convective systems.

3.  Scientific Value: The comparison highlights that both FYAI and CCIC act as robust, independent IWP datasets. Their combined use offers a unique opportunity for crossvalidation, which we believe will significantly benefit future scientific research on global cloud ice climatology.

Figure 4: It is impossible to make a sensible comparison to 2C-ICE. This comparison requires that MWHS and ERA5 IWPs along the 2C-ICE transects are extracted and plotted together with 2C-ICE IWP as a line plot (and the transects added to panels e and h). Please include information on what satellite that carried the two instruments considered.

Response: We appreciate your constructive feedback. We have redrawn this figure (now presented as Fig. 6 in the revised manuscript) following your suggestions to ensure a valid comparison. Furthermore, we have incorporated the CCIC dataset into the analysis to strengthen the cross-validation.

Line 305: The text reads as suggesting that Figure 5 shows that all IWP products exhibit fundamentally consistent spatial patterns. However, there are arguably larger differences in distribution between the FY-3X products than with the reference products, for example FY-3B and FY-3D. Text mentions that they are both "afternoon satellites". Are such differences expected?

Response: We appreciate this insight. The observed discrepancies between FY-3B and FY-3D are indeed expected, driven primarily by instrumental evolution rather than orbital timing.

Although both operate in afternoon orbits, FY-3D carries the advanced MWHS-II (featuring additional 89 and 118 GHz channels), which offers significantly higher sensitivity and structural detail than the first-generation MWHS-I on FY-3B. We have revised the manuscript to explicitly attribute these distributional differences to the superior detection capabilities of the second-generation instrument, correcting our previous oversimplification regarding spatial consistency.

Line 315: For clarity, please list which satellites are MWHS-I-based here again. To make it easier for users of the dataset to know what files to use.

Response: We appreciate your suggestion. We have revised the text to explicitly list the satellites based on MWHS-I in this section to ensure clarity for users.

Figure 7: According to the gridded dataset, FY-3B and FY-3C have overlapping data 2014–2019. The full FY-3B range should be included, or its omission should be motivated.

Response: In the updated figure, we have included the full temporal coverage for all satellites.

Line 330: Melia et al. (2016) does not seem to be a proper reference for DARDAR.

Response: We have corrected the reference to (Delanoë & Hogan, 2008).

Figure 7: A downward trend for IWP for the FY-3X satellites can be seen, which is not reflected in MODIS, VIIRS or ERA5. That should be discussed. The FY-3X retrievals also appear to have a more clear annual cycle than any other dataset, that also should be discussed.

Response: We appreciate your detailed analysis of the time series trends.

Regarding the downward trend, in the optimized version of the dataset, the FY-3X retrievals no longer exhibit a significant declining trend.

Regarding the annual cycle, we acknowledge that the FYAI product indeed displays a more distinct annual cycle and larger interannual variability compared to other datasets. We have added a discussion of this characteristic in the revised manuscript:

*"The time series of global total atmospheric ice mass reveals that the FYAI product exhibits larger interannual variability compared to the 2C-ICE baseline. This feature is non-uniform across the timeline, being most pronounced during the FY-3B era. While the variability in the later period decreases, the early record reflects the sensitivity differences inherent to the first-generation instrument."*

While we have characterized this behavior in the text, the precise physical or instrumental mechanisms driving this enhanced seasonality remain a subject for further investigation.

Sec 8: This section should be rewritten considering the general issues lifted above. In short, the section should also bring up limitations.

Response: We sincerely appreciate your valuable comments. We have substantially rewritten this section to address your concerns. The revised section follows a logical progression from product overview to critical evaluation, and finally to future outlook. Its primary focus is on transparency and user guidance; it rigorously details methodological trade-offs (e.g., precision vs. coverage), acknowledges specific instrument limitations (such as FY-3A underestimation and FY-3B aging effects), and provides explicit cautionary notes, ensuring users are well-informed about the dataset's appropriate application contexts before outlining a clear roadmap for future enhancements.

Line 348-353: In what way is the CLM a "distinct IWP product"?

Response: In the optimized version of the dataset, we have removed the CLM product, as it is essentially an intermediate by-product of the data processing pipeline.

Line 355-356: This reads as that neural networks automatically give a superior sampling. This is not correct, it is the choice of instrument that governs this aspect.

Response: We agree with your assessment that the sampling characteristics are determined by the instrument constellation rather than the network architecture. Consequently, we have removed this statement from the revised manuscript.

Line 357: The text indicates that "temporal continuity" has been achieved, while the main text mentions several data issues and biases between the FY-3X satellites are seen in Fig. 7.

Response: In the optimized dataset, we have successfully achieved "temporal continuity." Furthermore, to ensure transparency regarding data quality, we have explicitly tabulated the specific time periods where retrieval performance is suboptimal (as detailed in Table 3).

Line 386-393: This "outlook" is not very relevant and can be removed. If kept, it must be revised to properly account for ongoing work in these directions. In addition, the previous paragraph is already of outlook character.

Response: We sincerely appreciate your valuable comments. We have substantially rewritten this section to address your concerns.

Line 398: Make it very clear where to find the data. Consider putting it as the first sentence. "The presented datasets are available at". The current phrasing is unclear.

Response:The manuscript has been updated to reflect this comment.

Line 433-438: Duplicate reference.

Response: We have corrected this.

Data comments

In the gridded data, all FY3B_MWHSX_GBAL_L1_YYYY_MEAN.nc (iwp) files have clear

swath artefacts for certain months and they show up clearly when taking the yearly mean. Is there a suggested way to filter out these artefacts?

Response: We appreciate your keen observation. After a thorough investigation, we identified that the striping artifacts were caused by the inclusion of the "LandCover" feature in the model input. We have subsequently retrained the QRNN model without this feature, and the updated results confirm that the artifacts have been effectively eliminated.

Is there a recommended way to combine overlapping files from different satellites to get a best estimate for the monthly gridded IWP?

Response: We sincerely appreciate this valuable suggestion. Due to time constraints, we could not include this in the initial submission. However, we are currently processing a merged product and commit to uploading the optimized Level-3 (L3) monthly gridded IWP estimates within one month. These products will utilize a data-volume weighted approach to integrate observations during overlapping satellite periods.

There are some metadata issues in the NetCDF files; metadata units for IWP and CIWP are $kg/m^2$, but value ranges suggest they are in $g/m^2$. Cloud Mask Classification could benefit from a description on how to interpret the values.

Response: We apologize for the oversight regarding the metadata and any confusion it caused. In the optimized dataset, we have corrected the unit labels (to $g/m^2$) and have added detailed attribute descriptions for the cloud mask classification values to ensure clarity.

Due to a suspected technical issue with the data provider, download speeds are unusably slow (max 20 KB/s). It takes > 4 hours to download the 300 MB gridded level 3 data files, and makes it unfeasible to download the > 400 GB orbital zip file. We have tried to download this on 9th Oct, 10th Oct and 12th Oct from multiple different internet connections in an effort to rule out local technical issues. Due to this, we were unable to look at the orbital level 2 dataset and its usefulness for the scientific community appears limited. For this reason we feel forced, at this moment, to rate the data quality as poor.

References

Amell, A., Eriksson, P., & Pfreundschuh, S. (2022). Ice water path retrievals from Meteosat-9 using quantile regression neural networks. Atmospheric Measurement Techniques, 15(19), 5701-5717.

Amell, A., Pfreundschuh, S., & Eriksson, P. (2024). The chalmers cloud ice climatology: Retrieval implementation and validation. Atmospheric Measurement Techniques, 17(14), 4337-4368.

Duncan, D. I., & Eriksson, P. (2018). An update on global atmospheric ice estimates from satellite observations and reanalyses. Atmospheric Chemistry and Physics, 18(15), 11205-11219.

Ekelund, R., Eriksson, P., & Pfreundschuh, S. (2020). Using passive and active observations at microwave and sub-millimetre wavelengths to constrain ice particle models. Atmospheric Measurement Techniques, 13(2), 501-520.

Eliasson, S., Buehler, S. A., Milz, M., Eriksson, P., & John, V. O. (2011). Assessing observed and modelled spatial distributions of ice water path using satellite data. Atmospheric Chemistry and Physics, 11(1), 375-391.

Eriksson, P., Baró Pérez, A., Müller, N., Hallborn, H., May, E., Brath, M., ... & Ickes, L. (2025). Advancements and continued challenges in global modelling and observations of atmospheric ice masses. EGUsphere, 2025, 1-42.

Evans, K. F., & Stephens, G. L. (1995). Microweve radiative transfer through clouds composed of realistically shaped ice crystals. Part II. Remote sensing of ice clouds. Journal of Atmospheric Sciences, 52(11), 2058-2072.

Holl, G., Eliasson, S., Mendrok, J., & Buehler, S. A. (2014). SPARE-ICE: Synergistic ice water path from passive operational sensors. Journal of Geophysical Research: Atmospheres, 119(3), 1504-1523.

Li, J. L., Waliser, D. E., Chen, W. T., Guan, B., Kubar, T., Stephens, G., ... & Horowitz, L. (2012). An observationally based evaluation of cloud ice water in CMIP3 and CMIP5 GCMs and contemporary reanalyses using contemporary satellite data. Journal of Geophysical Research: Atmospheres, 117(D16).

Pfreundschuh, S., Kukulies, J., Amell, A., Hallborn, H., May, E., & Eriksson, P. (2025). The chalmers cloud ice climatology: A novel robust climate record of frozen cloud hydrometeor concentrations. Journal of Geophysical Research: Atmospheres, 130(6), e2024JD042618.

Zhao, L., & Weng, F. (2002). Retrieval of ice cloud parameters using the Advanced
Microwave Sounding Unit. Journal of Applied Meteorology, 41(4), 384-395.

Amell, A., Eriksson, P., & Pfreundschuh, S. (2022). Ice water path retrievals from Meteosat-9
using quantile regression neural networks. *Atmospheric Measurement Techniques*,

*15*(19), 5701–5717. https://doi.org/10.5194/amt-15-5701-2022

Amell, A., Pfreundschuh, S., & Eriksson, P. (2024). The Chalmers Cloud Ice Climatology:

Retrieval implementation and validation. *Atmospheric Measurement Techniques*,

*17*(14), 4337–4368. https://doi.org/10.5194/amt-17-4337-2024

An, N., Shang, H., Lesi, W., Ri, X., Shi, C., Tana, G., Bao, Y., Zheng, Z., Xu, N., Chen, L.,

Zhang, P., Ye, L., & Letu, H. (2023). A Cloud Detection Algorithm for Early

Morning Observations From the FY-3E Satellite. *IEEE Transactions on Geoscience

and Remote Sensing*, *61*, 1–15. https://doi.org/10.1109/TGRS.2023.3304985

Delanoë, J., & Hogan, R. J. (2008). A variational scheme for retrieving ice cloud properties

from combined radar, lidar, and infrared radiometer. *Journal of Geophysical

Research: Atmospheres*, *113*(D7), 2007JD009000.

https://doi.org/10.1029/2007JD009000

Deng, M., Mace, G. G., Wang, Z., & Lawson, R. P. (2013). Evaluation of Several A-Train Ice

Cloud Retrieval Products with In Situ Measurements Collected during the

SPARTICUS Campaign. *Journal of Applied Meteorology and Climatology*, *52*(4),

1014–1030. https://doi.org/10.1175/JAMC-D-12-054.1

Eriksson, P., Baró Pérez, A., Müller, N., Hallborn, H., May, E., Brath, M., Buehler, S. A., &

Ickes, L. (2025). *Advancements and continued challenges in global modelling and

observations of atmospheric ice masses*. Clouds and Precipitation/Remote

Sensing/Troposphere/Physics (physical properties and processes).

https://doi.org/10.5194/egusphere-2025-4634

Gerrit Holl, Holl, G., Eriksson, P., Salomon Eliasson, Eliasson, S., Jana Mendrok, Mendrok,

J., Stefan A. Buehler, & Buehler, S. (2014). SPARE-ICE: synergistic Ice Water Path

from passive operational sensors. *Journal of Geophysical Research*, *119*(3), 1504–

1523. https://doi.org/10.1002/2013jd020759

Gultepe, I., Heymsfield, A. J., Field, P. R., & Axisa, D. (2017). Ice-Phase Precipitation.

*Meteorological Monographs*, *58*, 6.1-6.36.

https://doi.org/10.1175/AMSMONOGRAPHS-D-16-0013.1

He, K., Zhang, X., Ren, S., & Sun, J. (2016). Deep Residual Learning for Image Recognition.

*2016 IEEE Conference on Computer Vision and Pattern Recognition (CVPR)*, 770–

778. https://doi.org/10.1109/CVPR.2016.90

Holl, G., Buehler, S. A., Rydberg, B., & Jiménez, C. (2010). Collocating satellite-based radar

and radiometer measurements – methodology and usage examples. *Atmospheric*

*Measurement Techniques*, *3*(3), Article 3. https://doi.org/10.5194/amt-3-693-2010

Holl, G., Eliasson, S., Mendrok, J., & Buehler, S. A. (2014). SPARE-ICE: Synergistic ice

water path from passive operational sensors. *Journal of Geophysical Research:*

*Atmospheres*, *119*(3), Article 3. https://doi.org/10.1002/2013JD020759

Li, J. -L. F., Waliser, D. E., Chen, W. -T., Guan, B., Kubar, T., Stephens, G., Ma, H. -Y.,

Deng, M., Donner, L., Seman, C., & Horowitz, L. (2012). An observationally based

evaluation of cloud ice water in CMIP3 and CMIP5 GCMs and contemporary

reanalyses using contemporary satellite data. *Journal of Geophysical Research:*

*Atmospheres*, *117*(D16), 2012JD017640. https://doi.org/10.1029/2012JD017640

Lundberg, S. M., & Lee, S.-I. (2017). *A Unified Approach to Interpreting Model Predictions*.

Tan, Z., Ma, S., Zhao, X., Yan, W., & Lu, W. (2019). Evaluation of Cloud Top Height

Retrievals from China's Next-Generation Geostationary Meteorological Satellite FY-4A. *Journal of Meteorological Research*, *33*(3), Article 3. https://doi.org/10.1007/s13351-019-8123-0

Tana, G., Lesi, W., Shang, H., Xu, J., Ji, D., Shi, J., Letu, H., & Shi, C. (2025). A New Cloud Water Path Retrieval Method Based on Geostationary Satellite Infrared Measurements. *IEEE Transactions on Geoscience and Remote Sensing*, *63*, 1–10. https://doi.org/10.1109/TGRS.2025.3526262

Vaswani, A., Shazeer, N., Parmar, N., Uszkoreit, J., Jones, L., Gomez, A. N., Kaiser, Ł. ukasz, & Polosukhin, I. (2017). Attention is All you Need. In I. Guyon, U. V. Luxburg, S. Bengio, H. Wallach, R. Fergus, S. Vishwanathan, & R. Garnett (Eds.), *Advances in Neural Information Processing Systems* (Vol. 30). Curran Associates, Inc. https://proceedings.neurips.cc/paper_files/paper/2017/file/3f5ee243547dee91fbd053c1c4a845aa-Paper.pdf

Wang, W., Wang, Z., He, Q., & Zhang, L. (2022). Retrieval of ice water path from the Microwave Humidity Sounder (MWHS) aboard FengYun-3B (FY-3B) satellite polarimetric measurements based on a deep neural network. *Atmospheric Measurement Techniques*, *15*(21), Article 21. https://doi.org/10.5194/amt-15-6489-2022

Wang, W., Xu, J., Letu, H., Zhang, L., Wang, Z., & Shi, J. (2024). A New Deep-Learning-Based Framework for Ice Water Path Retrieval From Microwave Humidity Sounder-II Aboard FengYun-3D Satellite. *IEEE Transactions on Geoscience and Remote*

*Sensing*, *62*, 1–14. https://doi.org/10.1109/TGRS.2024.3352654

RC2:

This manuscript presents a machine learning-based framework for retrieving the global Ice Water Path (IWP) from Fengyun series satellites (MWHS-I/II). The topic is of significant scientific importance and practical value, particularly as it develops a data product which is crucial for enriching the data sources available for global cloud and climate change research. The authors' effort to provide access to the data and some of the code aligns with the principles of open science and is commendable.

However, as a manuscript submitted to Earth System Science Data, the current version suffers from significant shortcomings in the completeness of the methodology, description of the data product, adequacy of the validation, and overall reproducibility. The manuscript currently reads more like a summary of a traditional research paper rather than a comprehensive dataset description. To meet the publication standards of ESSD, Major Revision is required. Below are my comments to improve the manuscript.

1. The core focus of ESSD is the data itself. I strongly recommend that the authors restructure the paper to shift the emphasis from "algorithm research" to a "dataset description." A dedicated section should be added to meticulously describe the final dataset's format, such as variables, spatio-temporal resolution, coverage, quality control flags.

   Response: We strongly agree with the reviewer that the core focus of ESSD should be the data itself.

   • Restructuring: We have significantly restructured the manuscript to shift the narrative focus from "algorithm development" to "dataset description," as suggested.

   • Dedicated Section: We added a dedicated section, section [Data Records], to comprehensively describe the final dataset. This section details the file format (NetCDF), variable definitions, spatiotemporal resolutions, global coverage, and quality control flags.

   • User Usability: We also included specific uncertainty information to ensure the dataset is transparent and convenient for user application.

2. Ambiguous Input Features: It is unclear which input features were used to retrieve the IWP. Was it only the brightness temperatures from the MWHS-I/II channels? Were other ancillary data used, such as viewing geometry (satellite zenith angle), surface type, or elevation? This information is essential for understanding the model's performance and for the study to be reproducible. A clear list of all input features is required.

Response: We thank the reviewer for emphasizing this point. We fully agree that a clear and comprehensive list of input features is vital for understanding the model's performance and ensuring reproducibility.

We apologize that this information, while included in the original Table, was not presented prominently enough.

To address this, we have created a new, dedicated table (Table 1) in the revised manuscript to explicitly list all input features used for the retrieval. As detailed in the new table.

3. Providing a per-pixel retrieval uncertainty estimate is a standard requirement for remote sensing data products. The authors need to state whether the dataset includes uncertainty information and, if so, explain how it was estimated (e.g., through model ensembling, quantile regression, or another method).

Response: We thank the reviewer for highlighting this standard requirement. We have updated the dataset to include per-pixel uncertainty estimates.

Method of Estimation: Leveraging the capability of our Quantile Regression Neural Network (QRNN), we explicitly output the conditional quantiles. We have selected the 5th and 95th percentiles to define the lower and upper bounds of the retrieval uncertainty, providing a 90% prediction interval for each pixel.

Data Availability: The updated dataset containing these uncertainty fields is currently being uploaded. We will provide the updated data access link (DOI/URL) in the revised manuscript.

4. There is ambiguity between "IWP" and "CIWP". The text mentions a process "employed to extract CIWP from the IWP data" (Line 121-122), yet the title and most of the text refer to "IWP." The authors need to clearly define whether the model retrieves the total atmospheric ice water path (IWP) or is specific to the cloud ice water path (CIWP). This distinction is fundamental to the dataset's definition and applicability.

Response: We thank the reviewer for this comment. To eliminate ambiguity, we have replaced the term "CIWP" with "Suspended Ice Water Path (SIWP)" throughout the revised manuscript. We have also added a clear definition in section [Collocations] stating that SIWP represents the cloud-only component (excluding precipitation), while the general "IWP" refers to the total ice water path.

5. The spatio-temporal collocation between the wide-swath Fengyun satellites and the nadir-viewing CloudSat is a critical step in building the training dataset. The authors have completely omitted a description of this process. What was the time window for a match? What was the spatial matching criterion (e.g., distance between the FY pixel center and the CloudSat footprint)? How many CloudSat profiles were averaged to match one FY pixel? These details must be added.

Response: We thank the reviewer for pointing this out. We apologize that these crucial details regarding the collocation process were not sufficiently prominent/clear in the original manuscript.

While these parameters (e.g., the 15-minute window and 7.5 km radius) were briefly mentioned in the original section [Collocations], we agree that they deserve a more detailed and explicit description to ensure reproducibility.

Therefore, in the revised manuscript, we have rewritten and highlighted this paragraph in section [Collocations] to clearly specify the matching criteria:

- Time Window: 15 minutes.

- Spatial Matching: Mean of CloudSat footprints within 7.5 km of the MWHS pixel center.

- Quality Control: Minimum of 9 CloudSat pixels with a Coefficient of Variation (CV) < 0.6.

Aside from the weaknesses of the overall motivation, there are several specific comments with the technical content:

1. (Line 120) "truth value": As mentioned in the major comments, please replace "truth value" with a more appropriate term like "reference value".

Response: We thank the reviewer for this valid point. We agree that "reference value" is scientifically more appropriate terminology than "true value" in this context. Accordingly, we have replaced all instances of "true value" (and "ground truth") with "reference value" throughout the revised manuscript.

2. (Line 121) CIWP Extraction Method: The phrase "employed to extract CIWP from the IWP data" is too vague. Please detail how the 2B-CLDCLASS data were used to derive CIWP from the 2C-ICE IWP. Was it by simply filtering for pixels classified as "high cloud"? How were multi-layer cloud scenarios handled?

Response: To isolate the Suspended Ice Water Path from the total IWP, we adopted the FLAG

methodology outlined by Li et al. (2012). This approach is based on a rigorous screening of the vertical cloud structure provided by the 2B-CLDCLASS product, rather than applying a simple filter for "high clouds."

As detailed in the revised section [Collocations], the filtering process follows strict criteria: all retrievals identified as surface precipitation are discarded. Furthermore, to minimize convective influence, data points classified as 'deep convection' or 'cumulus' in the 2B-CLDCLASS product are also excluded.

Regarding your specific question on multi-layer clouds: we do not exclude them by default. Instead, we rely on the vertical classification. Even in multi-layer scenarios, as long as the column does not contain the prohibited types (precipitation, deep convection, or cumulus), the total column IWP is retained and classified as SIWP.

3.  The introduction should more clearly articulate the unique advantages and necessity of developing an IWP product based on Fengyun satellites compared to existing products (e.g., from MODIS, AIRS, MLS). Does it fill a spatio-temporal gap, or does it offer potential accuracy improvements in specific areas?

    Response: Thank you for your constructive feedback. In the revised Introduction, we have clarified the unique contributions of the FYAI product compared to existing datasets (e.g., MODIS, AIRS, MLS) by highlighting several key aspects.

    First, the dataset incorporates observations from FY-3E, the world's first civil early-morning orbit meteorological satellite. This enables the Fengyun constellation to achieve global coverage every 4–6 hours, filling a critical gap in temporal sampling and allowing the capture of diurnal variations that are missed by single-orbit systems.

    Furthermore, unlike MODIS—which is limited to cloud-top observations—or MLS, which suffers from sparse horizontal sampling, FYAI provides day-and-night, wide-swath microwave measurements that can penetrate clouds. This offers a more complete view of the total IWP within large-scale cloud systems.

    A key innovation of FYAI is its separation of suspended ice from the total IWP. This directly addresses a common mismatch in model evaluation, as most general circulation models exclude precipitating ice, enabling more physically consistent comparisons between observations and simulations.

    Finally, FYAI serves as an independent, long-term climate data record, providing a valuable resource for cross-validation and uncertainty quantification within the global observational network.

4. Explain all abbreviations (e.g., FY-4A, IWP, MWHS) upon first appearance.

   Response: Thank you for your feedback. We have clarified all abbreviations upon their first mention in the text.

---

## Referee Report (RR1)

This study proposes and validates a global ice water path dataset (FYAI) based on passive microwave observations from the Fengyun-3 (FY-3) series satellites, spanning 15 years (2010–2024) and comprising two product levels: total ice water path and suspended ice water path. The study employs a machine learning framework and utilizes high-precision CloudSat/CALIPSO data as the training benchmark, demonstrating clear scientific significance and practical value. However, there are still some issues that need to be addressed.

1. It is recommended to supplement the introduction with research on microwave-based ice cloud remote sensing, both the advantages and disadvantages of microwave cloud remote sensing should also be explained.
2. It is recommended to add the novelty of the deep learning approach in the introduction.
3. Reference data and test data are different types of data. How can consistency be ensured between the two?
4. About the reference data,
   1) How does 2C-ICE obtain ice cloud parameters, what is the accuracy in distinguishing ice clouds from water clouds, and how is its penetration? Why can it be used as a reference for microwave ice cloud detection?
   2) 2C-ICE uses detection channels at 532 nm and 94 GHz, and there are differences in cloud detection sensitivity compared with the channels of Fengyun MWHS, such as sensitivity to ice cloud particles of different sizes and penetration through cloud layers. How can these differences explain the inconsistencies they cause?
   3) After 2011, CloudSat has no daytime data. How is the data trained then?

5. As validation data, CCIC and MODIS/VIIRS use visible-infrared optical data. How to explain the differences in sensitivity to ice cloud particles between these and microwave detection, since the cloud targets or cloud depths they observe are different? Although this was briefly mentioned in the comparison results below, how to address this difference requires countermeasures or explanations.
6. The part of 'quality control', it is recommended to express this paragraph concisely.
7. L192, what is the basis for choosing this threshold?
8. In the part of 'Data Record', the data format specifications and charts can be compressed or simplified, focusing more on their scientific value.
9. The payloads of Fengyun-3 from different batches, or in other words, the payloads of different satellites, do not necessarily have consistent radiometric baselines. It should first be explained how to ensure the consistency of Level 1 data across different satellite payloads. Although this is addressed in the uncertainty analysis later in the text, the issue of radiometric consistency of Level 1 data should be resolved before performing inversion. It would be helpful to look into the historical reprocessing data of Fengyun satellites, as this reprocessed data ensures the radiometric consistency of Level 1 data across historical datasets.
10. Figure 11, FYAI is consistent in magnitude with 2C-ICE and DARDAR over different times, but this does not prove that the two are consistent, because there may be temporal variations. Why not compare FYAI with the data from 2C-ICE that covers the same time period?

11. Some typos need to be noted. For example, L75, l145, Figure 'theblock', etc..